# Self-configuring feedback loops for sensorimotor control

**Sergio Oscar Verduzco-Flores\*, Erik De Schutter**

Computational Neuroscience Unit, Okinawa Institute of Science and Technology, Okinawa, Japan

**Abstract** How dynamic interactions between nervous system regions in mammals performs online motor control remains an unsolved problem. In this paper, we show that feedback control is a simple, yet powerful way to understand the neural dynamics of sensorimotor control. We make our case using a minimal model comprising spinal cord, sensory and motor cortex, coupled by long connections that are plastic. It succeeds in learning how to perform reaching movements of a planar arm with 6 muscles in several directions from scratch. The model satisfies biological plausibility constraints, like neural implementation, transmission delays, local synaptic learning and continuous online learning. Using differential Hebbian plasticity the model can go from motor babbling to reaching arbitrary targets in less than 10 min of in silico time. Moreover, independently of the learning mechanism, properly configured feedback control has many emergent properties: neural populations in motor cortex show directional tuning and oscillatory dynamics, the spinal cord creates convergent force fields that add linearly, and movements are ataxic (as in a motor system without a cerebellum).

## Editor's evaluation

This solid modelling study presents a valuable contribution toward understanding the neural control of movement. The authors show that a minimal model comprising key sensorimotor cortical areas as well as a spinal circuits controlling a limb readily replicates landmark observations from behavioural and electrophysiological studies. This work will be of broad interest to motor control researchers, as well as to neurophysiologists interested in testing the predictions derived from this model.

**\*For correspondence:**
sergio.verduzco@gmail.com

**Competing interest:** The authors declare that no competing interests exist.

## Introduction

### The challenge

Neuroscience has made great progress in decoding how cortical regions perform specific brain functions like primate vision (**Kaas and Collins, 2003**; **Ballard and Zhang, 2021** and rodent navigation **Chersi and Burgess, 2015**; **Moser et al., 2017**). Conversely, the evolutionary much older motor control system still poses fundamental questions, despite a large body of experimental work. This is because, in mammals, in addition to areas in cortex like premotor and motor areas and to some degree sensory and parietal ones, many *extracortical regions* have important and unique functions: basal ganglia, thalamus, cerebellum, pons, brain stem nuclei like the red nucleus and spinal cord (**Eccles, 1981**; **Loeb and Tsianos, 2015**). These structures are highly interconnected by fast conducting axons and all show strong dynamic activity changes, related to the ongoing dynamics of the performed motor act. Clinical and lesion studies have confirmed the necessity of each of these regions for normal smooth motor control of arm reaching (**Shadmehr and Wise, 2005**; **Arber and Costa, 2018**).

Fully understanding motor control will thus entail understanding the simultaneous function and interplay of all brain regions involved. Little by little, new experimental techniques will allow us to

monitor more neurons, in more regions, and for longer periods (*Tanaka et al., 2018*, e.g.). But to make sense of these data computational models must step up to the task of integrating all those regions to create a functional neuronal machine.

Finally, relatively little is known about the neural basis of motor development *in infants* (*Hadders-Algra, 2018*). Nevertheless, a full understanding of primate motor control will not only require explanation of how these brain regions complement and interact with each other but also how this can be learned during childhood.

With these challenges in mind we recently developed a motor control framework based on differential Hebbian learning (*Verduzco-Flores et al., 2022*). A common theme in physiology is the control of *homeostatic variables* (e.g. blood glucose levels, body temperature, etc.) using negative feedback mechanisms (*Woods and Ramsay, 2007*). From a broad perspective, our approach considers the musculoskeletal system as an extension of this homeostatic control system: movement aims to make

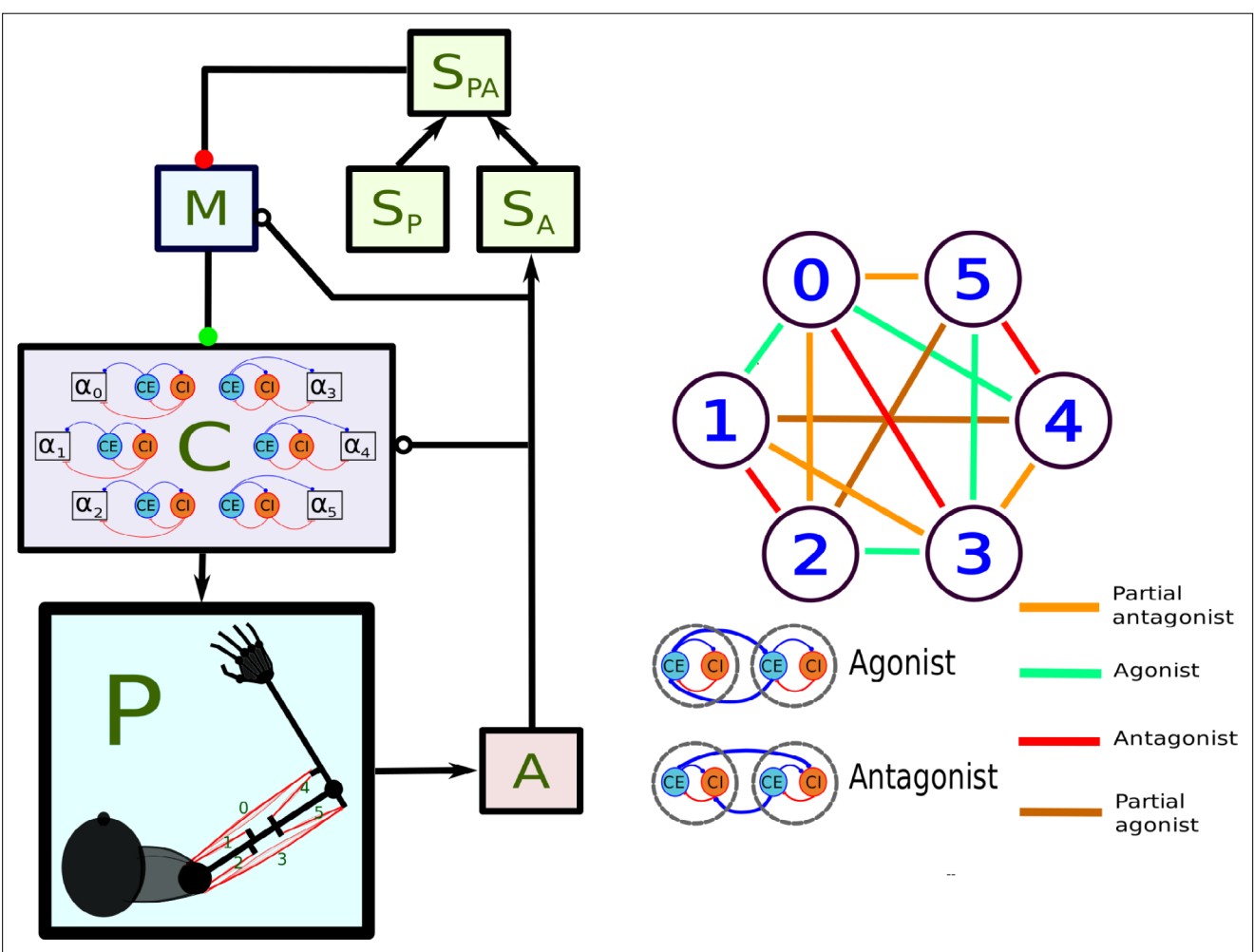

**Figure 1.** Main components of the model. In the left panel, each box stands for a neural population, except for P, which represents the arm and the muscles. Arrows indicate static connections, open circles show *input correlation* synapses, and the two colored circles show possible locations of synapses with the learning rule in *Verduzco-Flores et al., 2022*. In the *spinal learning* model the green circle connections are plastic, and the red circle connections are static. In the *cortical learning* model the red circle connections are plastic, whereas the green circle connections are static. In the *static network* all connections are static. A : afferent population. $S_A$ : Somatosensory cortex, modulated by afferent input. $S_P$ : somatosensory cortex, prescribed pattern. $S_{PA}$ : population signaling the difference between $S_P$ and $S_A$ : primary motor cortex. C : spinal cord. Inside the C box the circles represent the excitatory (**CE**) and inhibitory (**CI**) interneurons, organized into six pairs. The interneurons in each pair innervate an alpha motoneuron ($\alpha$), each of which stimulates one of the six muscles in the arm, numbered from 0 to 5. The trios consisting of **CE**, **CI**, $\alpha$ units are organized into agonists and antagonists, depending on whether their $\alpha$ motoneurons cause torques in similar or opposite directions. These relations are shown in the right-side panel.

the external environment conducive to the internal control of homeostatic variables (e.g. by finding food, or shelter from the sun).

Our working hypothesis (see *Verduzco-Flores et al., 2022*) is that control of homeostatic variables requires a feedback controller that uses the muscles to produce a desired set of sensory perceptions. The motosensory loop, minimally containing motor cortex, spinal cord, and sensory cortex may implement that feedback controller. To test this hypothesis we implemented a relatively complete model of the sensorimotor loop (*Figure 1*), using the learning rules in *Verduzco-Flores et al., 2022* to produce 2D arm reaching. The activity of the neural populations and the movements they produced showed remarkable consistency with the experimental observations that we describe next.

## Relevant findings in motor control

Before describing our modeling approach, we summarize some of the relevant experimental data that will be important to understanding the results. We focus on three related issues: (1) the role of the spinal cord in movement, (2) the nature of representations in motor cortex, and (3) muscle synergies, and how the right pattern of muscle activity is produced.

For animals to move, spinal motoneurons must activate the skeletal muscles. In general, descending signals from the corticospinal tract do not activate the motoneurons directly, but instead provide input to a network of excitatory and inhibitory interneurons (*Bizzi et al., 2000*; *Lemon, 2008*; *Arber, 2012*; *Asante and Martin, 2013*; *Alstermark and Isa, 2012*; *Jankowska, 2013*; *Wang et al., 2017*; *Ueno et al., 2018*). Learning even simple behaviors involves long-term plasticity, both at the spinal cord (SC) circuit, and at higher regions of the motor hierarchy (*Wolpaw et al., 1983*; *Grau, 2014*; *Meyer-Lohmann et al., 1986*; *Wolpaw, 1997*; *Norton and Wolpaw, 2018*). Despite its obvious importance, there are comparatively few attempts to elucidate the nature of the SC computations, and the role of synaptic plasticity.

The role ascribed to SC is closely related to the role assumed from motor cortex, particularly M1. One classic result is that M1 pyramidal neurons of macaques activate preferentially when the hand is moving in a particular direction. When the preferred directions of a large population of neurons are added as vectors, a population vector appears, which points close to the hand's direction of motion (*Georgopoulos et al., 1982*; *Georgopoulos et al., 1986*). This launched the hypothesis that M1 represents kinematic, or other high-level parameters of the movement, which are transformed into movements in concert with the SC. This hypothesis mainly competes with the view that M1 represents muscle forces. Much research has been devoted to this issue (*Kakei et al., 1999*; *Truccolo et al., 2008*; *Kalaska, 2009*; *Georgopoulos and Stefanis, 2007*; *Harrison and Murphy, 2012*; *Tanaka, 2016*; *Morrow and Miller, 2003*; *Todorov, 2000*, e.g.).

Another important observation is that the preferred directions of motor neurons cluster around one main axis. As shown in *Scott et al., 2001*, this suggests that M1 is mainly concerned with dynamical aspects of the movement, rather than representing its kinematics.

A related observation is that the preferred directions in M1 neurons experience random drifts that overlap learned changes (*Rokni et al., 2007*; *Padoa-Schioppa et al., 2004*). This leads to the hypothesis that M1 is a redundant network that is constantly using feedback error signals to capture the *task-relevant dimensions*, placing the configuration of synaptic weights in an *optimal manifold*.

A different perspective for studying motor cortex is to focus on how it can produce movements, rather than describing its activity (*Shenoy et al., 2013*). One specific proposal is that motor cortex has a collection of pattern generators, and specific movements can be created by combining their activity (*Shenoy et al., 2013*; *Sussillo et al., 2015*). Experimental support for this hypothesis came through the surprising finding of rotational dynamics in motor cortex activity (*Churchland et al., 2012*), suggesting that oscillators with different frequencies are used to produce desired patterns. This begs the question of how the animal chooses its desired patterns of motion.

Selecting a given pattern of muscle activation requires *planning*. Motor units are the final actuators in the motor system, but they number in the tens of thousands, so planning movements in this space is unfeasible. A low-dimensional representation of desired limb configurations (such as the location of the hand in Euclidean coordinates) is better. Movement generation likely involves a coordinate transformation, from the endpoint coordinates (e.g. hand coordinates) into actuator coordinates (e.g. muscle lengths), from which motor unit activation follows directly. Even using pure engineering methods, as for robot control, computing this coordinate transformation is very challenging.

For example, this must overcome kinematic redundancies, as when many configurations of muscle lengths put the hand in the same location.

The issue of coordinate transformation is central for motor control (*Shadmehr and Wise, 2005*; *Schöner et al., 2018*; *Valero-Cuevas, 2009*; *motor primitives and muscle synergies* are key concepts in this discussion). Representing things as combinations of elementary components is a fundamental theme in applied mathematics. For example, linear combinations of basis vectors can represent any vector, and linear combinations of wavelets can approximate any smooth function (*Keener, 1995*). In motor control, this idea arises in the form of motor primitives. Motor primitives constitute a set of basic motions, such that that any movement can be decomposed into them (*Giszter, 2015*; *Mussa–Ivaldi and Bizzi, 2000*; *Bizzi et al., 1991*). This is closely related to the concept of synergies. The term 'synergy' may mean several things (*Kelso, 2009*; *Bruton and O'Dwyer, 2018*), but in this paper, we use it to denote a pattern of muscle activity arising as a coherent unit. Synergies may be composed of motor primitives, or they may be the motor primitives themselves.

A promising candidate for motor primitives comes in the form of convergent force fields, which have been observed for the hindlimbs of frogs and rats (*Giszter et al., 1993*; *Mussa-Ivaldi et al., 1994*, or in the forelimbs of monkeys *Yaron et al., 2020*). In experiments where the limb is held at a particular location, local stimulation of the spinal cord will cause a force to the limb's endpoint. The collection of these force vectors for all of the limb endpoint's positions forms a force field, and these force fields have two important characteristics: (1) they have a unique fixed point and (2) simultaneous stimulation of two spinal cord locations produces a force field which is the sum of the force fields from stimulating the two locations independently. It is argued that movement planning may be done in terms of force fields, since they can produce movements that are resistant to perturbations, and also permit a solution to the problem of coordinate transformation with redundant actuators (*Mussa–Ivaldi and Bizzi, 2000*).

The neural origin of synergies, and whether they are used by the motor system is a matter of ongoing debate (*Tresch and Jarc, 2009*; *de Rugy et al., 2013*; *Bizzi and Cheung, 2013*). To us, it is of interest that single spinal units found in the mouse (*Levine et al., 2014* and monkey *Takei et al., 2017*) spinal cord (sometimes called Motor Synergy Encoders, or MSEs) can reliably produce specific patterns of motoneuron activation.

## Model concepts

We believe that it is impossible to understand the complex dynamical system in biological motor control without the help of computational modeling. Therefore, we set out to build a minimal model that could eventually control an autonomous agent, while still satisfying biological plausibility constraints.

Design principles and biological-plausibility constraints for neural network modeling have been proposed before (*Pulvermüller et al., 2021*; *O'Reilly, 1998*; *Richards et al., 2019*). Placing emphasis on the motor system, we compiled a set of characteristics that cover the majority of these constraints. Namely:

- Spanning the whole sensorimotor loop.
- Using only neural elements. Learning their connection strengths is part of the model.
- Learning does not rely on a training dataset. It is instead done by synaptic elements using local information.
- Learning arises from continuous-time interaction with a continuous-space environment.
- There is a clear vision on how the model integrates with the rest of the brain in order to enact more general behavior.

Our aim is hierarchical control of homeostatic variables, with the spinal cord and motor cortex at the bottom of this hierarchy. At first glance, spinal plasticity poses a conundrum, because it changes the effect of corticospinal inputs. Cortex is playing a piano that keeps changing its tuning. A solution comes when we consider the corticospinal loop (e.g. the long-loop reflex) as a negative control system, where the spinal cord activates the effectors to reduce an error. The role of cortex is to produce perceptual variables that are controllable, and can eventually improve homeostatic regulation. In this regard, our model is a variation of Perceptual Control Theory (*Powers, 1973*; *Powers, 2005*), but if the desired value of the controller is viewed as a prediction, then this approach resembles

active inference models (*Adams et al., 2013*). Either way, the goal of the system is to reduce the difference between the desired and the perceived value of some variable.

If cortex creates representations for perceptual variables, the sensorimotor loop must be configured so those variables can be controlled. This happens when the error in those variables activates the muscles in a way that brings the perceived value closer to the desired value. In other words, we must find the input-output structure of the feedback controller implicit in the long-loop reflex. We have found that this important problem can be solved by the differential Hebbian learning rules introduced in *Verduzco-Flores et al., 2022*. We favor the hypothesis that this learning takes place is in the connections from motor cortex to interneurons and brainstem. Nevertheless, we show that all our results are valid if learning happens in the connections from sensory to motor cortex.

In the Results section we will describe our model, its variations, and how it can learn to reach. Next we will show that many phenomena described above are present in this model. These phenomena emerge from having a properly configured neural feedback controller with a sufficient degree of biological realism. This means that even if the synaptic weights of the connections are set by hand and are static, the phenomena still emerge, as long as the system is configured to reduce errors. In short, we show that a wealth of phenomena in motor control can be explained simply by feedback control in the sensorimotor loop, and that this feedback control can be configured in a flexible manner by the learning rules presented in *Verduzco-Flores et al., 2022*.

## Results

### A neural architecture for motor control

The model in this paper contains the main elements of the long-loop reflex, applied to the control of a planar arm using six muscles. The left panel of *Figure 1* shows the architecture of the model, which contains 74 firing rate neurons organized in six populations. This architecture resembles a feedback controller that makes the activity in a neural population $S_A$ approach the activity in a different population $S_P$.

The six firing-rate neurons (called *units* in this paper) in $S_A$ represent a region of somatosensory cortex, and its inputs consist of the static gamma (II) afferents. In steady state, activity of the II afferents is monotonically related to muscle length (*Mileusnic et al., 2006*), which in turn can be used to prescribe hand location. Other afferent signals are not provided to $S_A$ in the interest of simplicity.

$S_P$ represents a different cortical layer of the same somatosensory region as $S_A$, where a 'desired' or 'predicted' activity has been caused by brain regions not represented in the model. Each firing rate neuron in $S_A$ has a corresponding unit in $S_P$, and they represent the mean activity at different levels of the same microcolumn (*Mountcastle, 1997*). $S_{PA}$ is a region (either in sensory or motor cortex) that conveys the difference between activities in $S_P$ and $S_A$, which is the error signal to be minimized by negative feedback control.

Population $A$ represents sensory thalamus and dorsal parts of the spinal cord. It contains 18 units with logarithmic activation functions, each receiving an input from a muscle afferent. Each muscle provides proprioceptive feedback from models of the Ia, Ib, and II afferents. In rough terms, Ia afferents provide information about contraction velocity, and Ib afferents signal the amount of tension in the muscle and tendons.

Population $M$ represents motor cortex. Ascending inputs to $M$ arise from population $A$, and use a variation of the *input correlation* learning rule (*Porr and Wörgötter, 2006*), where the $S_{PA}$ inputs act as a learning signal. The input correlation rule enhances the stability of the controller. More details are presented in Methods. The $S_{PA}$ inputs to $M$ can either be static, or use a learning rule to be described below.

To represent positive and negative values, both $M$ and $S_{PA}$ use a 'dual representation', where each error signal is represented by two units. Let $e_i = s_P^i - s_A^i$ be the error associated with the $i$-th muscle. One of the two $S_{PA}$ units representing $e_i$ is a monotonic function of $\max(e_i, 0)$, whereas the other unit increases according to $\max(-e_i, 0)$. These opposing inputs, along with mutual inhibition between the two units creates dynamics where sensorimotor events cause both excitatory and inhibitory responses, which agrees with experimental observations (*Shafi et al., 2007*; *Steinmetz et al., 2019*; *Najafi et al., 2020*), and allows transmitting 'negative' values using excitatory projections. Dual units in $M$ receive the same inputs, but with the opposite sign.

Plasticity mechanisms within the sensorimotor loop should specify which muscles contract in order to reduce an error signaled by $S_{PA}$. We suggest that this plasticity could take place in the spinal cord and/or motor cortex. To show that our learning mechanisms work regardless of where the learning takes place, we created two main configurations of the model. In the first configuration, called the 'spinal learning' model, a 'spinal' network $C$ transforms the $M$ outputs into muscle stimulation. $C$ learns to transform sensory errors into appropriate motor commands using a differential Hebbian learning rule (*Verduzco-Flores et al., 2022*). In this configuration, the error input to each $M$ unit comes from one of the $S_{PA}$ activities. A second configuration, called the 'cortical learning' model, has 'all-to-all' connections from $S_{PA}$ to $M$ using the differential Hebbian rule, whereas the connections from $M$ to $C$ use appropriately patterned static connections. Both configurations are basically the same model; the difference is that one configuration has our learning rule on the inputs to $C$, whereas the other has it on the inputs to $M$ (*Figure 1*).

While analyzing our model we reproduced several experimental phenomena (described below). Interestingly, these phenomena did not arise because of the learning rules. To make this explicit, we created a third configuration of our model, called the 'static network'. This configuration does not change the weight of any synaptic connection during the simulation. The initial weights were hand-set to approximate the optimal solution everywhere (see Methods). We will show that all emergent phenomena in the paper are also present in the static network.

We explain the idea behind the differential Hebbian rule as applied in the connections from $M$ to $C$. $C$ contains $N$ interneurons, whose activity vector we denote as $\mathbf{c} = [c_1, \ldots, c_N]$. The input to each of these units is an $M$ dimensional vector $\mathbf{e} = [e_1, \ldots, e_M]$. Each unit in $C$ has an output $c_i = \sigma\left(\sum_j \omega_{ij} e_j\right)$, where $\sigma(\cdot)$ is a positive sigmoidal function. The inputs are assumed to be errors, and to reduce them we want $e_j$ to activate $c_i$ when $c_i$ can reduce $e_j$. One way this could happen is when the weight $\omega_{ij}$ from $e_j$ to $c_i$ is proportional to the negative of their sensitivity derivative:

$$\omega_{ij} \propto -\frac{\partial e_j}{\partial c_i}. \tag{1}$$

Assuming a monotonic relation between the motor commands and the errors, relation 1 entails that errors will trigger an action to cancel them, with some caveats considered in *Verduzco-Flores et al., 2022*. Synaptic weights akin to *Equation 1* can be obtained using a learning rule that extracts correlations between the derivatives of $c_i$ and $e_j$ (see Methods). Using this rule, the commands coming from population $C$ can eventually move the arm so that $S_A$ activity resembles $S_P$ activity.

$C$ is organized to capture the most basic motifs of spinal cord connectivity using a network where balance between excitation and inhibition is crucial (*Berg et al., 2007*; *Berg et al., 2019*; *Goulding et al., 2014*). Each one of six $\alpha$ motoneurons stimulate one muscle, and is stimulated by one excitatory ($CE$), and one inhibitory ($CI$) interneuron. $CE$ and $CI$ stimulate one another, resembling the classic Wilson-Cowan model (*Cowan et al., 2016*). The trios composed of $\alpha$, $CE$, and $CI$ neurons compose a group that controls the activation of one muscle, with $CE$ and $CI$ receiving convergent inputs from $M$. This resembles the premotor network model in *Petersen et al., 2014*. ($\alpha$, $CE$, $CI$) trios are connected to other trios following the agonist-antagonist motif that is common in the spinal cord (*Pierrot-Deseilligny and Burke, 2005*). This means that $CE$ units project to the $CE$ units of agonists, and to the $CI$ units of antagonists (*Figure 1*, right panel). When the agonist/antagonist relation is not strongly defined, muscles can be 'partial'aASaS agonists/antagonists, or unrelated.

Connections from $A$ to $C$ (the 'short-loop reflex') use the input correlation learning rule, analogous to the connections from $A$ to $M$.

Direct connections from $M$ to alpha motoneurons are not necessary for the model to reach, but they were introduced in new versions because in higher primates these connections are present for distal joints (*Lemon, 2008*). Considering that bidirectional plasticity has been observed in corticomotoneural connections (*Nishimura et al., 2013*), we chose to endow them with the differential Hebbian rule of *Verduzco-Flores et al., 2022*.

Because timing is essential to support the conclusions of this paper, every connection has a transmission delay, and all firing rate neurons are modeled with ordinary differential equations.

All the results in this paper apply to the three configurations described above (spinal learning, cortical learning, and static network). To emphasize the robustness and potential of the learning mechanisms, in the Appendix we introduce two variations of the spinal learning model (in the *Variations of*

*the spinal learning model* section). All results in the paper also apply to those two variations. In one of the variations (the 'synergistic' network), each spinal motoneuron stimulates two muscles rather than one. In the second variation (the 'mixed errors' network), the inputs from $S_{PA}$ to $M$ are not one-to-one, but instead come from a matrix that combines multiple error signals as the input to each $M$ unit.

Since most results apply to all configurations, and since results could depend on the random initial weights, we report simulation results using three means and three standard deviations ($m_1 \pm \sigma_1 | m_2 \pm \sigma_2 | m_3 \pm \sigma_3$), with the understanding that these three value pairs correspond respectively to the spinal learning, motor learning, and static network models. The statistics come from 20 independent simulations with different initial conditions.

A reference section in the Appendix (the *Comparison of the 5 configurations* section) summarizes the basic traits of all different model configurations (including the two variations of the spinal learning model), and compiles all their numerical results.

For each configuration, a single simulation was used to produce all the representative plots in different sections of the paper.

## The model can reach by matching perceived and desired sensory activity

Reaches are performed by specifying an $S_P$ pattern equal to the $S_A$ activity when the hand is at the target. The acquisition of these $S_P$ patterns is not in the scope of this paper (but see *Verduzco-Flores et al., 2022*).

We created a set of random targets by sampling uniformly from the space of joint angles. Using this to set a different pattern in $S_P$ every 40 s, we allowed the arm to move freely during 16 $S_P$ target presentations. To encourage exploratory movements we used noise and two additional units described in the Methods.

All model configurations were capable of reaching. To decide if reaching was learned in a trial we took the average distance between the hand and the target (the *average error*) during the last four target presentations. Learning was achieved when this error was smaller than 10 cm.

The system learned to reach in 99 out of 100 trials (20 for each configuration). One simulation with the spinal learning model had an average error of 14 cm during the last 4 reaches of training. To assess the speed of learning we recorded the average number of target presentations required before the error became less than 10 cm for the first time. This average number of failed reaches before the first success was: ($1.8 \pm 2 | 1.2 \pm .9 | 0 \pm 0$).

*Figure 2A* shows the error through 16 successive reaches (640 s of in silico time) in a typical case for the spinal learning model. A supplementary video (*Appendix 1—Video 1*) shows the arm's movements during this simulation. Figures similar to *Figure 2* can be seen for all configurations as figure supplements (*Figure 2—figure supplement 1*) (*Figure 2—figure supplement 2*).

In *Figure 2A*, the error increases each time a new target was presented (yellow vertical lines), but as learning continues it was consistently reduced below 10 cm.

Panel B also shows the effect of learning, as the hand's Cartesian coordinates eventually track the target coordinates whenever they change. This is also reflected as the activity in $S_A$ becoming similar to the activity in $S_P$ (panel C).

Panels D and E of *Figure 2* show the activity of a few units in population $M$ and population $C$ during the 640 s of this training phase. During the first few reaches, $M$ shows a large imbalance between the activity of units and their duals, reflecting larger errors. Eventually these activities balance out, leading to a more homogeneous activity that may increase when a new target appears. M1 activation patterns that produce no movement are called the *null-space activity* (*Kaufman et al., 2014*). In our case, this includes patterns where $M$ units have the same activity as their duals. This, together with the noise and oscillations intrinsic to the system cause the activity in $M$ and $C$ to never disappear.

In panel E, the noise in the $C$ units becomes evident. It can also be seen that inhibition dominates excitation (due to $CE$ to $CI$ connections), which promotes stability in the circuit.

We tested whether any of the novel elements in the model were superfluous. To this end, we removed each of the elements individually and checked if the model could still learn to reach. In conclusion, removing individual elements generally deteriorated performance, but the factor that proved essential for all configurations with plasticity was the differential Hebbian learning in the

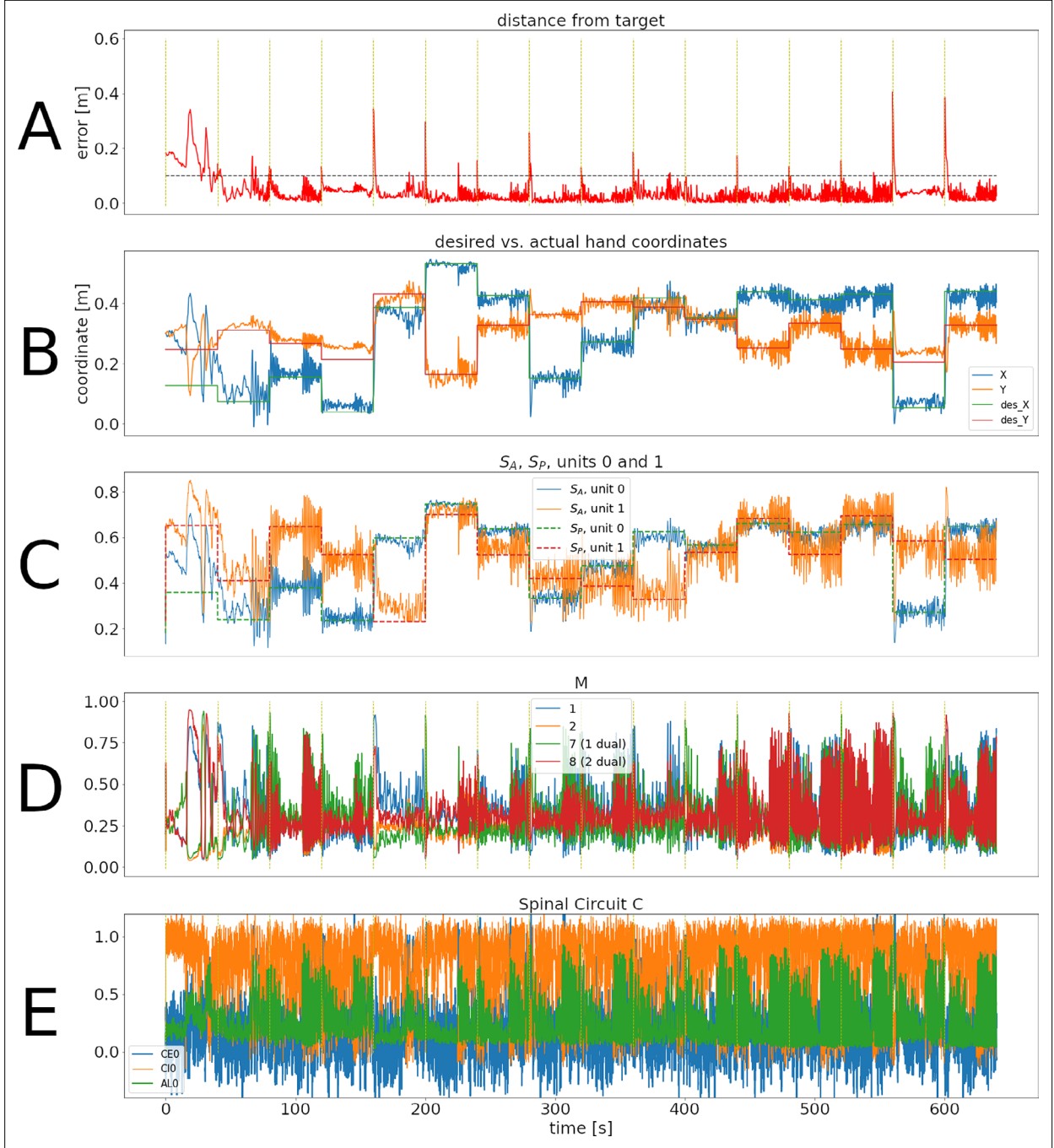

**Figure 2.** Representative training phase of a simulation for the spinal learning model. (**A**) Distance between the target and the hand through 640 s of simulation, corresponding to 16 reaches to different targets. The horizontal dotted line corresponds to 10 cm. The times when $S_P$ changes are indicated with a vertical, dotted yellow line. Notice that the horizontal time axis is the same for all panels of this figure. The average error can be seen to decrease through the first two reaches. (**B**) Desired versus actual hand coordinates through the training phase. The straight lines denote the desired X (green) and Y (red) coordinates of the hand. The noisy orange and blue lines show the actual coordinates of the hand. (**C**) Activity of units 0 and 1 in $S_P$ and $S_A$. This panel shows that the desired values in the $S_P$ units (straight dotted lines) start to become tracked by the perceived values. (**D**) Activity of $M$ units 1, 2, and their duals. Notice that even when the error is close to zero the activity in the $M$ units does not disappear. E: Activity of the $CE, CI, \alpha$ trio for muscle 0. The intrinsic noise in the units causes ongoing activity. Moreover, the inhibitory activity (orange line) dominates the excitatory activity (blue line).

The online version of this article includes the following figure supplement(s) for figure 2:

**Figure supplement 1.** Representative training phase of the simulation for the cortical learning configuration.

**Figure supplement 2.** Representative training phase of the simulation for the static network.

*Figure 2 continued on next page*

connections from $M$ to $C$ or from $S_{PA}$ to $M$. For details, see the the Appendix section titled *The model fails when elements are removed*.

## Center-out reaching 1: The reach trajectories present traits of cerebellar ataxia

In order to compare our model with experimental data, after the training phase we began a standard center-out reaching task. Switching to this task merely consisted of presenting the targets in a different way, but for the sake of smoother trajectories we removed the noise from the units in $C$ or $M$.

*Figure 3A* shows the eight peripheral targets around a hand rest position. Before reaching a peripheral target, a reach to the center target was performed, so the whole experiment was a single continuous simulation controlled by the $S_P$ pattern.

Peripheral targets were selected at random, each appearing six times. This produced 48 reaches (without counting reaches to the center), each one lasting 5 s. Panels B through D of *Figure 3* show the trajectories followed by the hand in the three configurations. During these 48 reaches the average distance between the hand and the target was $(3.3 \pm .01 | 2.9 \pm .001 | 2.9 \pm .0003)$ centimeters.

Currently our system has neither cerebellum nor visual information. Lacking a 'healthy' model to make quantitative comparisons, we analyzed and compared them to data from cerebellar patients.

For the sake of stability and simplicity, our system is configured to perform slow movements. Fast and slow reaches are different in cerebellar patients (*Bastian et al., 1996*). Slow reaches undershoot the target, follow longer hand paths, and show movement decomposition (joints appear to move one at a time). In *Figure 3* the trajectories begin close to the 135 degree axis, indicating a slower response at the elbow joint. With the parameters used, the spinal learning and cortical learning models tend to undershoot the target, whereas in the static network the hand can oscillate around the target.

The traits of the trajectories can be affected by many hyperparameters in the model, but the dominant factor seems to be the gain in the control loop. Our model involves delays, activation latencies, momentum, and interaction torques. Unsurprisingly, increasing the gain leads to oscillations along with faster reaching. On the other hand, low gain leads to slow, stable reaching that often undershoots the target. Since we do not have a cerebellum to overcome this trade off, the gain was the only hyperparameter that was manually adjusted for all configurations (See Methods). In particular, we adjusted the slope of the $M$ and $S_A$ units so the system was stable, but close to the onset of oscillations. Gain was allowed to be a bit larger in the static network so oscillations could be observed. The figure supplements for *Figure 3* shows more examples of configurations with higher gain (See *Gain and oscillations* in Appendix 1 for details).

The shape of the trajectory also depends on the target. Different reach directions cause different interaction forces, and encounter different levels of viscoelastic resistance from the muscles.

*Figure 4* reveals that the approach to the target is initially fast, but gradually slows down. Healthy subjects usually present a bell-shaped velocity profile, with some symmetry between acceleration and deceleration. This symmetry is lost with cerebellar ataxia (*Becker et al., 1991*; *Gilman et al., 1976*).

We are not aware of center-out reaching studies for cerebellar patients in the dark, but (*Day et al., 1998*) does examine reaching in these conditions. Summarizing its findings:

1. Movements were slow.
2. The endpoints had small variation, but they had constant errors.
3. Longer, more circuitous trajectories, with most changes in direction during the last quarter.
4. Trajectories to the same target showed variations.

From *Figures 3 and 4* we can observe constant endpoint errors when the gain is low, in the spinal and cortical learning models. Circuitous trajectories with a pronounced turn around the end of the third quarter are also observed. Individual trajectories can present variations. A higher gain, as in the static network on the right plots, can increase these variations, as illustrated in the figure supplements for Appendix 1.

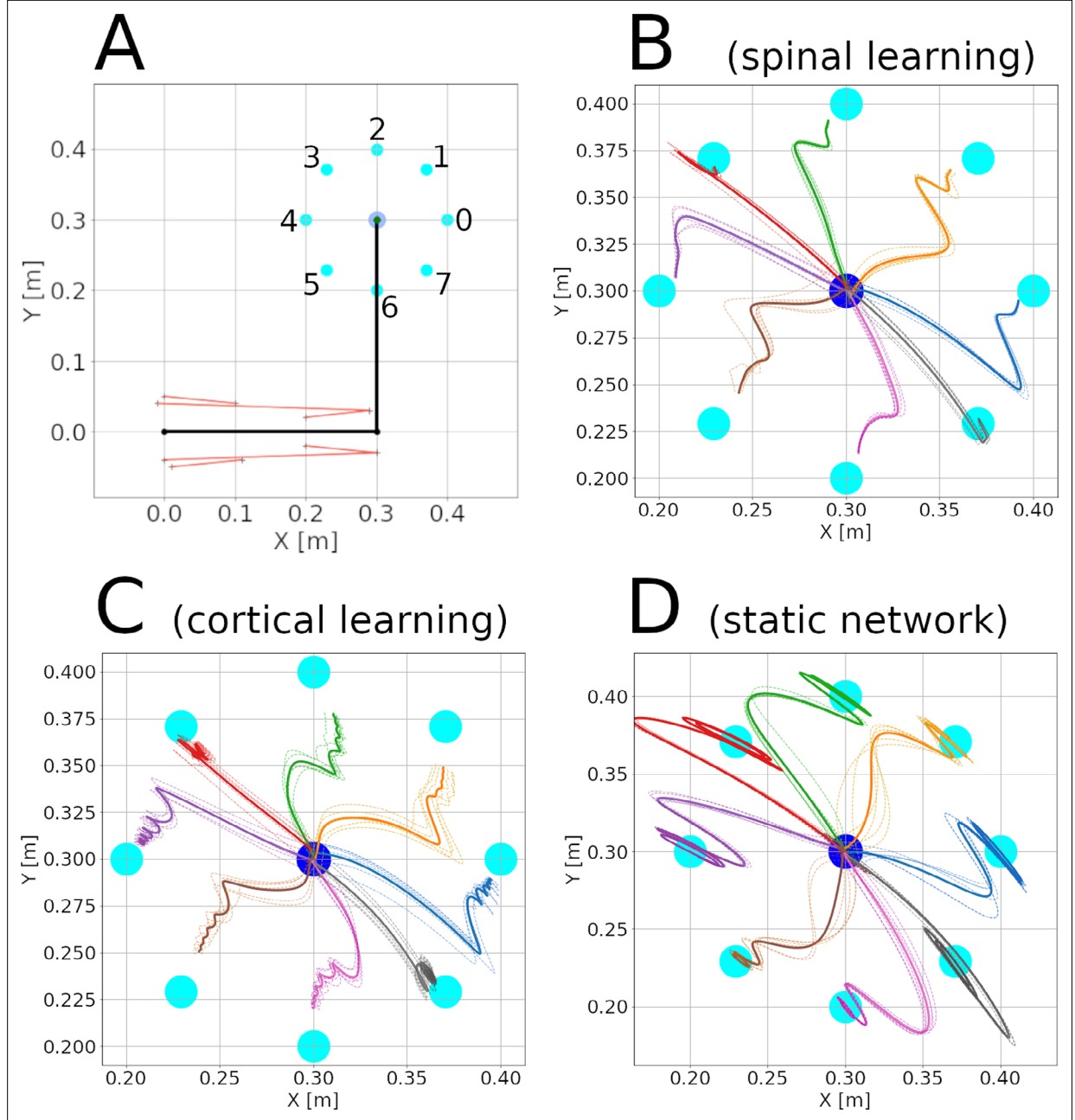

**Figure 3.** Center-out reaching. (**A**) The arm at its resting position, with hand coordinates (0.3, 0.3) meters, where a center target is located. Eight peripheral targets (cyan dots) were located on a circle around the center target, with a 10 cm radius. The muscle lines, connecting the muscle insertion points, are shown in red. The shoulder is at the origin, whereas the elbow has coordinates (0.3, 0). Shoulder insertion points remain fixed. (**B-F**) Hand trajectories for all reaches in the three configurations. The trajectory's color indicates the target. Dotted lines show individual reaches, whereas thick lines indicate the average of the 6 reaches.

The online version of this article includes the following figure supplement(s) for figure 3:

**Figure supplement 1.** Center-out reaching before the control loop gain was adjusted.

**Figure supplement 2.** Training phase of the simulation for the static network before gain was reduced.

**Figure supplement 3.** Training phase of the simulation for the synergistic network before gain was reduced.

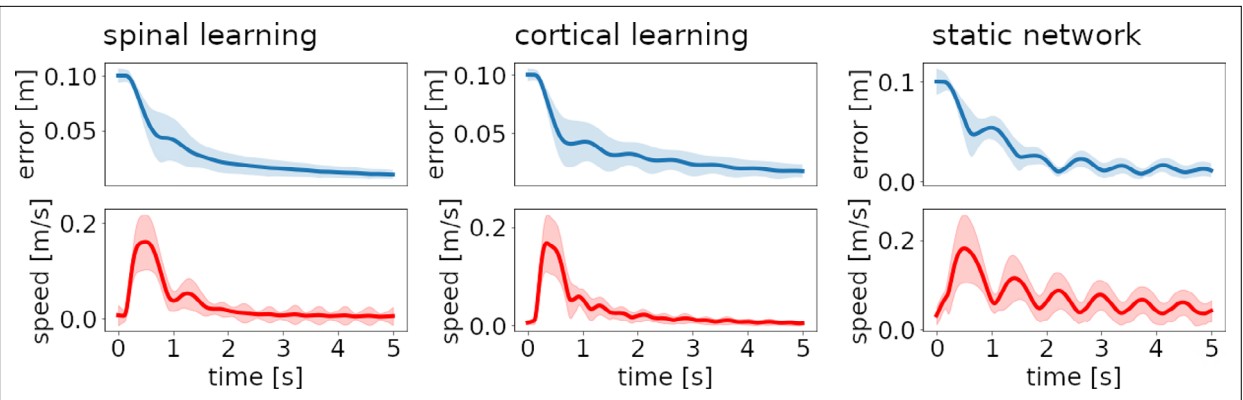

**Figure 4.** Distance to target and reach velocity through time for the three configurations. Thick lines show the average over 48 reaches (8 targets, 6 repetitions). Filled stripes show standard deviation. For the spinal and cortical learning configurations (left and center plots) the hand initially moves quickly to the target, but the direction is biased, so it needs to gradually correct the error from this initial fast approach; most of the variance in error and velocity appears when these corrections cause small-amplitude oscillations. In the case of the static network (right plots) oscillations are ongoing, leading to a large variance in velocity.

## Center-out reaching 2: Directional tuning and preferred directions

To find whether directional tuning could arise during learning, we analyzed the $M$ population activity for the 48 radial reaches described in the previous subsection.

For each of the 12 units in $M$, *Figure 5A* shows the mean firing rate of the unit when reaching each of the 8 targets. The red arrows show the Preferred Direction (PD) vectors that arise from these distributions of firing rates. For the sake of exposition, *Figure 5* shows data for the simpler case of one-to-one connectivity between $S_{PA}$ and $M$ in the spinal learning model, but these results generalize to the case when each $M$ unit receives a linear combination of the $S_{PA}$ activities (the 'mixed errors' variation presented in the *Variations of the spinal learning model* section of the Appendix.)

We found that $(11.8 \pm .4|12 \pm 0|12 \pm 0)$ units were significantly tuned to reach direction ($p < 0.001$, bootstrap test), with PD vectors of various lengths. The direction of the PD vectors is not mysterious. Each $M$ unit controls the length error of one muscle. *Figure 5B* shows that the required contraction length depends on both the target and the muscle. The PD vectors of units 0–5 point to the targets that require the most contraction of their muscle. Units 6–11 are the duals of 0–5, and their PD is in the opposite direction. *Figure 5C* shows that the PD may also be inferred from the muscle activity, reflected as average tension.

In the case when each $M$ unit receives a linear combination of $S_{PA}$ errors, its PD can be predicted using a linear combination of the 'directions of maximum contraction' shown in *Figure 5B*, using the same weights as the $S_{PA}$ inputs. When accounting for the length of the PD vectors, this can predict the PD angle with a coefficient of determination $R^2 \approx (.74 \pm .18|.88 \pm .14|.86 \pm .01)$.

As mentioned in the Introduction, the PDs of motor cortex neurons tend to align in particular directions *Scott et al., 2001*. This is almost trivially true for this model, since the PD vectors are mainly produced by linear combinations of the vectors in *Figure 5B*.

*Figure 6* shows the PD for all the $M$ units in a representative simulation for each of the configurations. In every simulation, the PD distribution showed significant bimodality ($p < 0.001$). The main axis of the PD distribution (see Methods) was $(59 \pm 7|52 \pm 2|54 \pm .5)$ degrees.

To compare with (*Scott et al., 2001*) we rotate this line 45 degrees so the targets are in the same position relative to the shoulder (e.g. *Lillicrap and Scott, 2013 Figure 1*, *Kurtzer et al., 2006 Figure 1*). This places the average main axes above in a range between 99 and 104 degrees, comparable to the 117 degrees in *Scott et al., 2001*.

The study in *Lillicrap and Scott, 2013* suggested that a rudimentary spinal cord feedback system should be used to understand *why* the PD distribution arises. Our model is the first to achieve this.

The PD vectors are not stationary, but experience random fluctuations that become more pronounced in new environments (*Rokni et al., 2007*; *Padoa-Schioppa et al., 2004*). The brain is constantly remodeling itself, without losing the ability to perform its critical operations (*Chambers and Rumpel, 2017*). Our model is continuously learning, so we tested the change in the PDs by

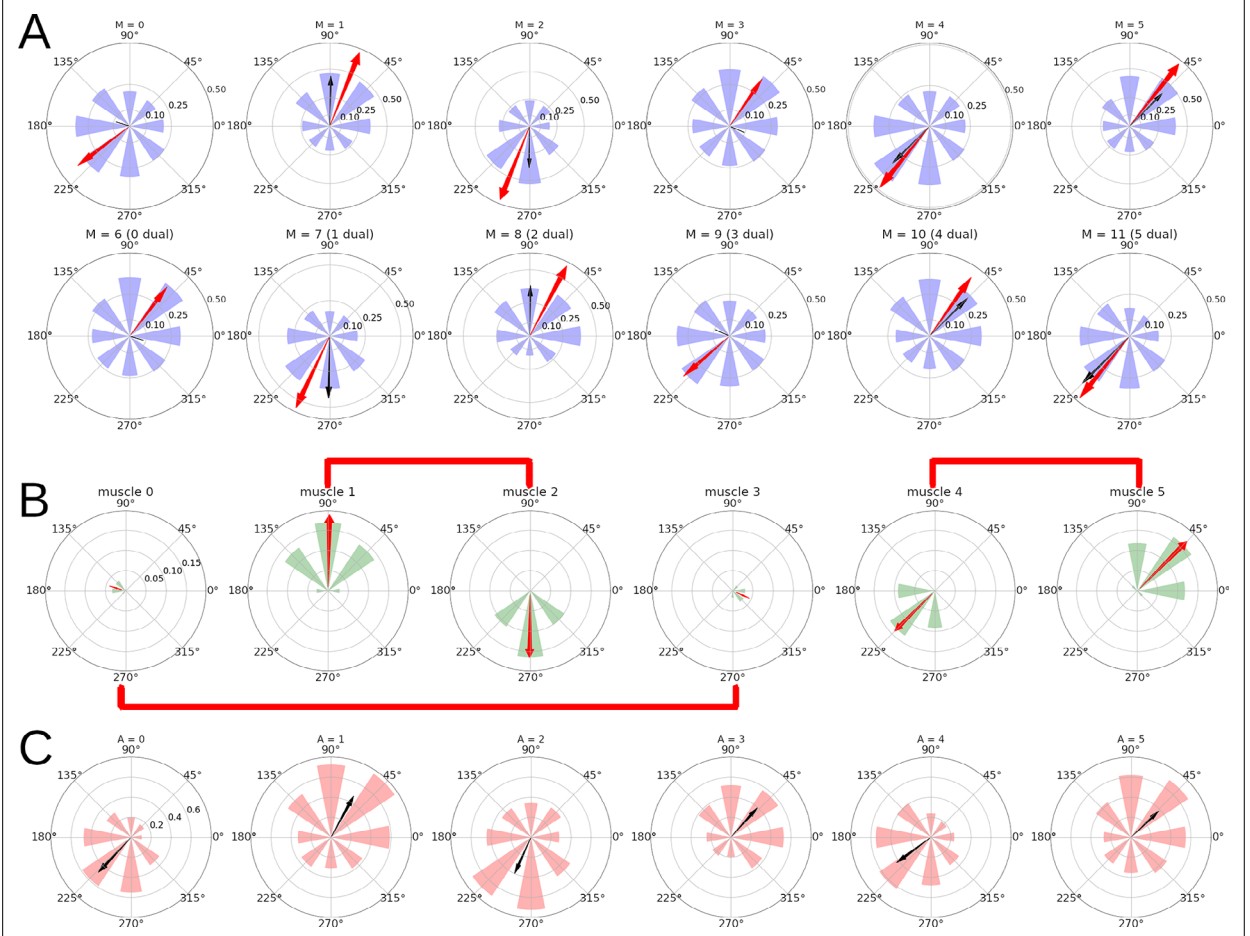

**Figure 5.** Directional tuning of the units in $M$ for a simulation with the spinal learning model. (**A**) Average firing rate per target, and preferred direction (see Methods) for each of the 12 units in $M$. Each polar plot corresponds to a single unit, and each of the 8 purple wedges corresponds to one of the 8 targets. The length of a wedge indicates the mean firing rate when the hand was reaching the corresponding target. The red arrow indicates the direction and relative magnitude of the PD vector. The black arrow shows the predicted PD vector, in this case just the corresponding arrows from panel B. (**B**) For each muscle and target, a wedge shows the muscle's length at rest position minus the length at the target, divided by the rest position length. The red arrow comes from the sum of the wedges taken as vectors, and represents the muscle's direction of maximum contraction. Plots corresponding to antagonist muscles are connected by red lines. (**C**) Average activity of the 6 $A$ units indicating muscle tension. The black arrows come from the sum of wedges taken as vectors, showing the relation between muscle tension and preferred direction.

setting 40 additional center-out reaches (no intrinsic noise) after the previous experiment, once for each configuration.

To encourage changes we set 10 different targets instead of 8. After a single trial for each configuration the change in angle for the 12 PD vectors had means and standard deviations of $(3.3 \pm 2.4 | 4.9 \pm 2.1 | .3 \pm .2)$ degrees. Larger changes (around 7 degrees) could be observed in the 'mixed errors' variation of the model, presented in the Appendix (*Variations of the spinal learning model* section). We also measured the change in the preferred directions of the muscles, obtained as in *Figure 5C*. This yielded differences and standard deviations $(3.8 \pm 2.1 | 6.4 \pm 2.9 | .2 \pm .2)$ degrees.

The average distance between hand and target during the 40 reaches was $(3 | 3.6 | 2.9)$ cm, showing that the hand was still moving towards the targets, although with different errors due to their new locations.

## Center-out reaching 3: Rotational dynamics

Using a dynamical systems perspective, (*Shenoy et al., 2013*) considers that the muscle activity $\mathbf{m}(t)$ (a vector function of time) arises from the cortical activity vector $\mathbf{r}(t)$ after it is transformed by the downstream circuitry:

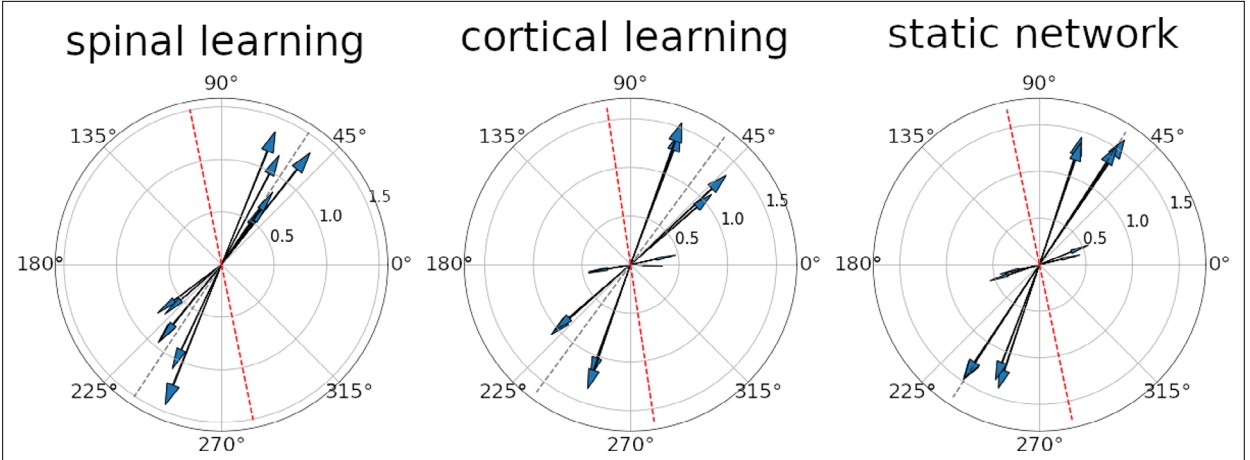

**Figure 6.** Preferred direction vectors for the 12 $M$ units. In all three plots the arrows denote the direction and magnitude of the preferred direction (PD) for an individual unit. The gray dotted lines shows the main axis of the distribution. The red dotted lines are a 45 degree rotation of the gray line, for comparison with *Scott et al., 2001*. It can be seen that all configurations display a strong bimodality, especially when considering the units with a larger PD vector. The axis where the PD vectors tend to aggregate is in roughly the same position for the three configurations.

$$\mathbf{m}(t) = G[\mathbf{r}(t)]. \qquad (2)$$

It is considered that the mapping $G[\cdot]$ may consist of sophisticated controllers, but for the sake of simplicity this mapping is considered static, omitting spinal cord plasticity. The cortical activity arises from a dynamical system:

$$\tau\dot{\mathbf{r}}(t) = h(\mathbf{r}(t)) + \mathbf{u}(t), \qquad (3)$$

where $\mathbf{u}(t)$ represents inputs to motor cortex from other areas, and $h(\cdot)$ is a function that describes how the state of the system evolves.

A difficulty associated with *Equation 3* is explaining how $\mathbf{r}(t)$ generates a desired muscle pattern $\mathbf{m}(t)$ when the function $h(\cdot)$ represents the dynamics of a recurrent neural network. One possibility is that M1 has intrinsic oscillators of various frequencies, and they combine their outputs to shape the desired pattern. This prompted the search for oscillatory activity in M1 while macaques performed center-out reaching motions. A brief oscillation (in the order of 200ms, or 5 Hz) was indeed found in the population activity (*Churchland et al., 2012*, and the model in *Sussillo et al., 2015*) was able to reproduce this result, although this was done in the open-loop version of *Equations 2 and 3*, where $\mathbf{u}(t)$ contains no afferent feedback (this is further commented in the Supplemental Discussion).

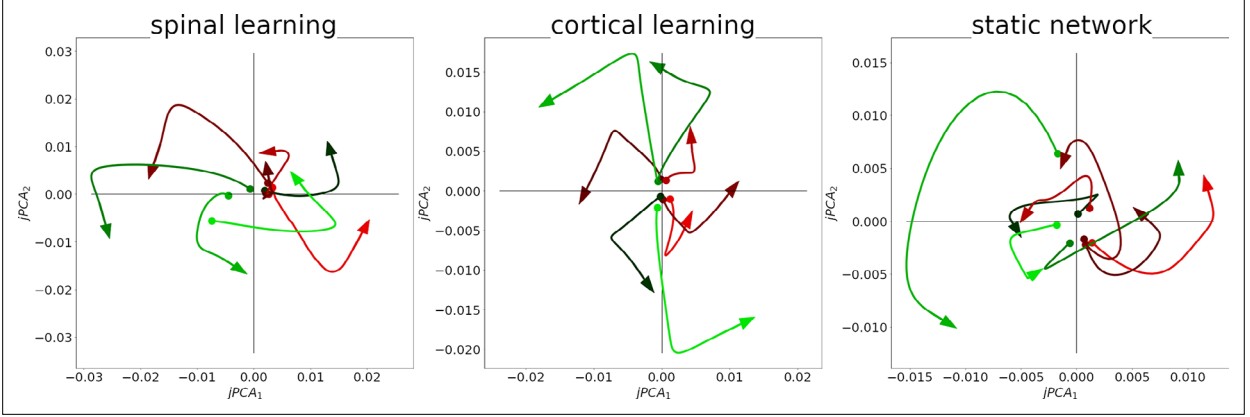

**Figure 7.** Rotational dynamics in the M population in a representative simulation for all configurations. Each plot shows the first two jPCA components during 0.25 s, for each of the 8 conditions/targets. Traces are colored according to the magnitude of their initial $jPCA_1$ component, from smallest (green) to largest (red).

Recently it was shown that the oscillations in motor cortex can arise when considering the full sensorimotor loop, without the need of recurrent connections in motor cortex (*Kalidindi et al., 2021*). A natural question is whether our model can also reproduce the oscillations in *Churchland et al., 2012* without requiring M1 oscillators or recurrent connections.

The analysis in *Churchland et al., 2012* is centered around measuring the amount of rotation in the M1 population activity. The first step is to project the M1 activity vectors onto their first six principal components. These six components are then rotated so the evolution of the activity maximally resembles a pure rotation. These rotated components are called the 'jPCA vectors'. The amount of variance in the M1 activity explained by the first two jPCA vectors is a measure of rotation. The Methods section provides more details of this procedure.

Considering that we have a low-dimensional, non-spiking, slow-reaching model, we can only expect to qualitatively replicate the essential result in *Churchland et al., 2012*, which is most of the variance being contained in the first jPCA plane.

We replicated the jPCA analysis, with adjustments to account for the smaller number of neurons, the slower dynamics, and the fact that there is no delay period before the reach (See Methods). The result can be observed in *Figure 7*, where 8 trajectories are seen in the plots. Each trajectory is the average activity of the 12 $M$ units when reaching to one of the 8 targets, projected onto the jPCA plane. The signature of a rotational structure in these plots is that most trajectories circulate in a counterclockwise direction. Quantitatively, the first jPCA plane (out of six) captures ($.42 \pm .04 | .42 \pm .04 | .46 \pm .03$) of the variance.

With this analysis we show that our model does not require intrinsic oscillations in motor cortex to produce rotational dynamics, in agreement with (*Kalidindi et al., 2021* and *DeWolf et al., 2016*).

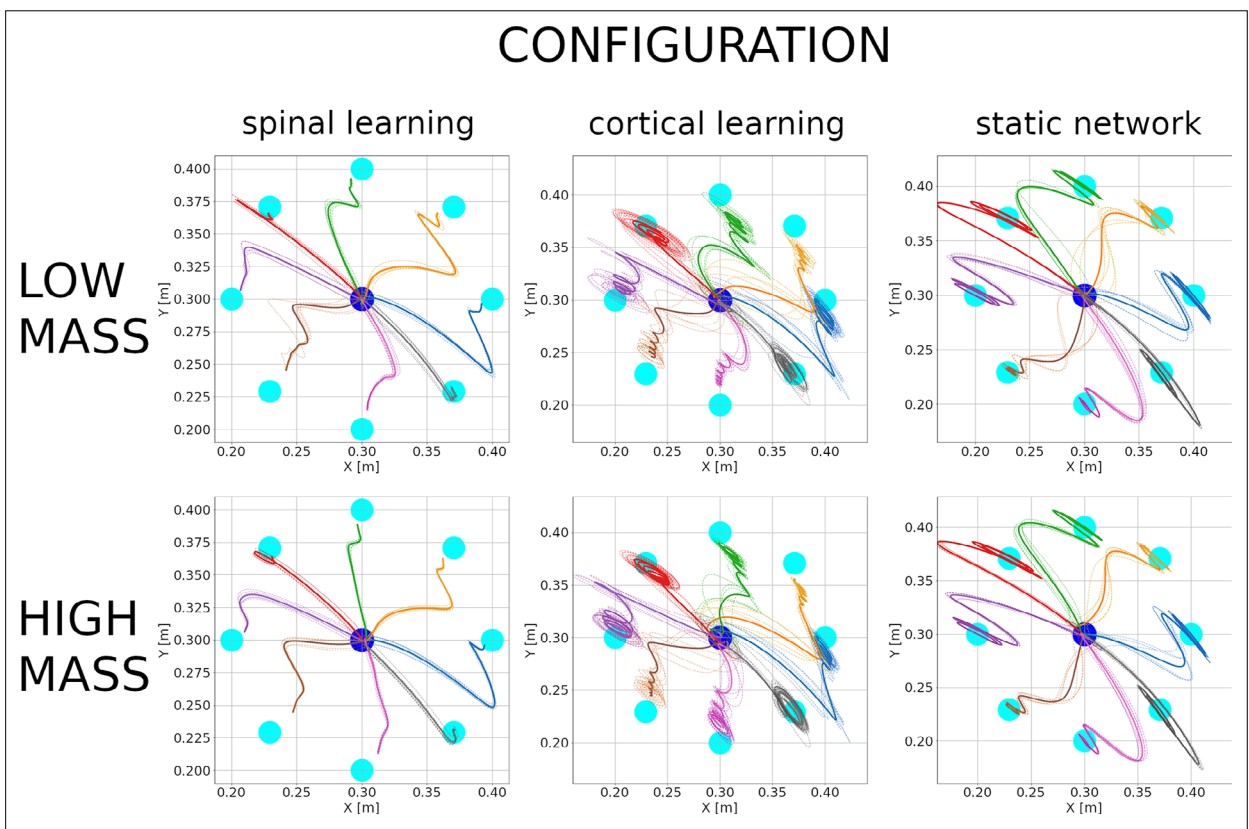

**Figure 8.** Hand trajectories with low mass (0.8 kg, top 3 plots) and high mass (1.2 kg, bottom 3 plots) for the 3 configurations. Plots are as in *Figure 3*. The spinal learning model and the static network show qualitatively similar trajectories compared to those in *Figure 3*. In contrast, the cortical learning model began to display considerable endpoint oscillations for several targets after its mass was reduced. These oscillations persist after the mass has been increased.

## The effect of changing the mass

Physical properties of the arm can change, not only as the arm grows, but also when tools or new environments come into play. As a quick test of whether the properties in this paper are robust to moderate changes, we changed the mass of the arm and forearm from 1 to 0.8 kg and ran one simulation for each of the five configurations.

With a lighter arm the average errors during center-out reaching were (2.5|3.2|3) cm. The hand trajectories with a reduced mass can be seen in the top 3 plots of *Figure 8*. We can observe that the spinal learning model slightly reduced its mean error, whereas the cortical learning model increased it. This can be understood by noticing that a reduction in mass is akin to an increase in gain. The spinal learning model with its original gain was below the threshold of oscillations at the endpoint, and a slight mass decrease did not change this. The cortical learning model with the original gain was already oscillating slightly, and an increase in gain increased the oscillations.

In the same simulation, after the center-out reaching was completed, we once more modified the mass of the arm and forearm, from 0.8 to 1.2 kg, after which we began the center-out reaching again. This time the center-out reaching errors were (2.4|3.3|2.9) cm. The hand trajectories for this high mass condition are in the bottom 3 plots in *Figure 8*. It can be seen that the spinal learning and cortical learning models retained their respectively improved and decreased performance, whereas the static network performed roughly the same for all mass conditions. A tentative explanation is that with reduced mass the synaptic learning rules tried to compensate for faster movements with weights that effectively increased the gain in the loop. After the mass was increased these weights did not immediately revert, leading to similar trajectories after the increase in mass.

The results of the paper still held after our mass manipulations. For all configurations, PD vectors could be predicted with a coefficient of determination between .74 and .92; All units in $M$ were significantly tuned to direction; the main axis of the PD distribution ranged between 56 and 61 degrees, and the first jPCA plane captured between 33% and 58% of the variance.

## Spinal stimulation produces convergent direction fields

Due to the viscoelastic properties of the muscles, the mechanical system without active muscle contraction will have a fixed point with lowest potential energy at the arm's rest position. Limited amounts of muscle contraction shift the position of that fixed point. This led us to question whether this could produce convergent force fields, which as discussed before are candidate motor primitives, and have been found experimentally.

To simulate local stimulation of an isolated spinal cord we removed all neuronal populations except for those in $C$, and applied inputs to the individual pairs of $CE, CI$ units projecting to the same motoneuron. Doing this for different starting positions of the hand, and recording its initial direction of motion, produces a *direction field*. A direction field maps each initial hand location to a vector pointing in the average direction of the force that initially moves the hand.

The first two panels of *Figure 9* show the result of stimulating individual E-I pairs in $C$, which will indeed produce direction fields with different fixed points.

We found that these direction fields add approximately linearly (*Figure 9D*). More precisely, let $D(a + b)$ be the direction field from stimulating spinal locations $a$ and $b$ simultaneously, and $\alpha_{a+b}(x, y)$ be the angle of $D(a + b)$ at hand coordinates $(x, y)$. Using similar definitions for $D(a), D(b), \alpha_a(x, y), \alpha_b(x, y)$, we say the direction fields add linearly if $\alpha_{a+b}(x, y) = \alpha_a(x, y) + \alpha_b(x, y), \ \forall (x, y)$.
We define the mean angle difference between $D(a + b)$ and $D(a) + D(b)$ as

$$\gamma_{a,b} = \sum_{x,y} \frac{\alpha_{a+b}(x, y) - \left( \alpha_a(x, y) + \alpha_b(x, y) \right)}{N_s}, \tag{4}$$

where $N_s$ is the number of $(x, y)$ sample points. We found that when averaged over the 15 ($C_1$) or 144 ($C_2$) possible $(a, b)$ pairs, the mean of $\gamma_{a,b}$ was 13.5 degrees.

Randomly choosing two possibly different pairs $(a, b)$ and $(c, d)$ for the stimulation locations leads to a mean angle difference of 37.6 degrees between the fields $D(a + b)$ and $D(c) + D(d)$. A bootstrap test showed that these angles are significantly larger ($p < 0.0001$) than in the previous case where $(a, b) = (c, d)$.

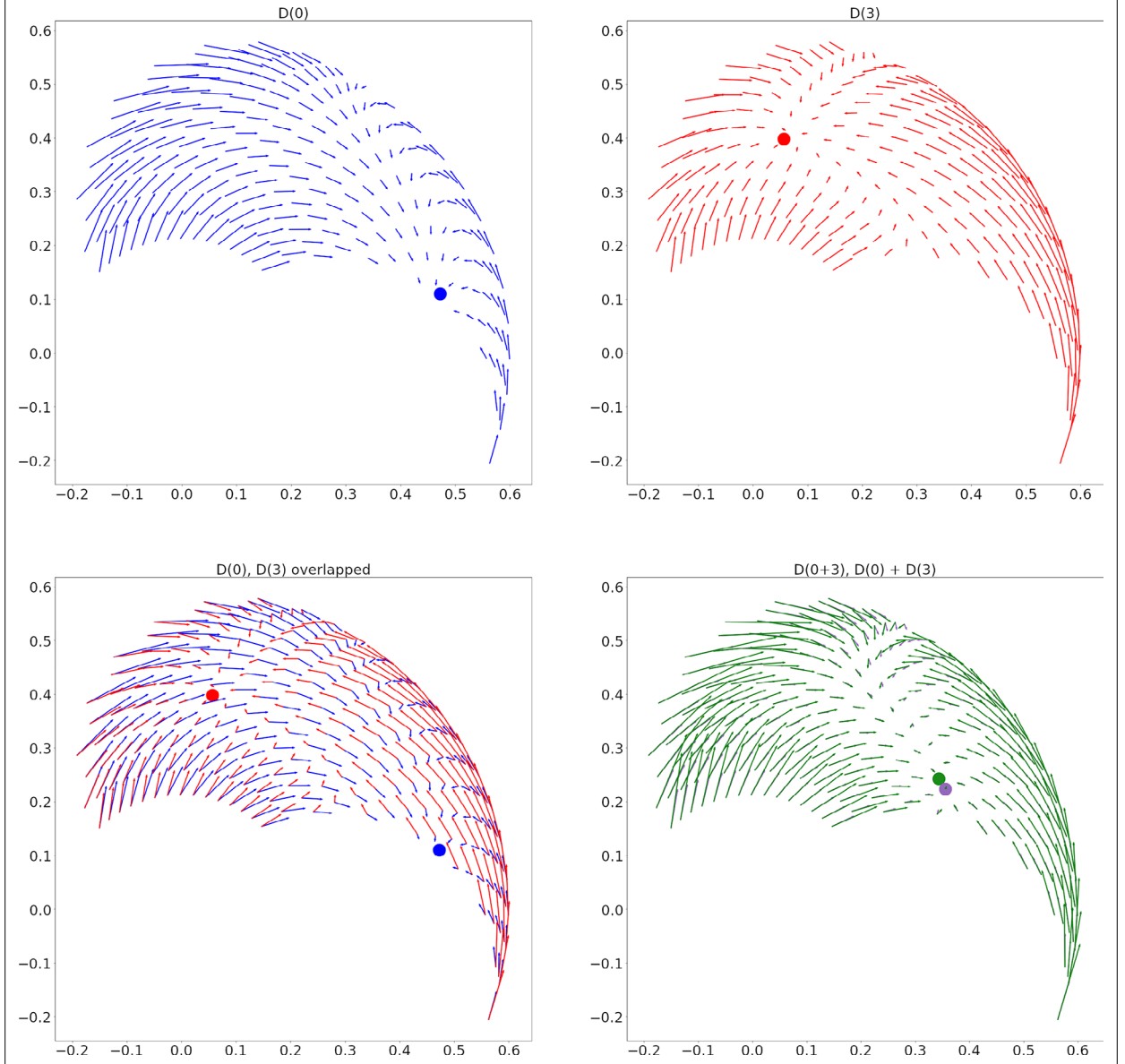

**Figure 9.** Two sample direction fields and their linear addition for circuit $C_1$. (**A**) Direction Field (DF) from stimulation of the interneurons for muscle 0 (biarticular biceps). The approximate location of the fixed point is shown with a blue dot. (**B**) DF from stimulation of muscle 3 (biarticular triceps) interneurons. A red dot shows the fixed point. (**C**) Panels A and B overlapped. (**D**) In green, the DF from stimulating the interneurons for muscles 0 and 3 together. In purple, the sum of the DFs from panels A and B. Dots show the fixed points. The average angle between the green and purple vectors is 4 degrees.

The resting field is defined as the direction field when no units are stimulated. Removing the resting field from $D(a + b), D(a)$, and $D(b)$ does not alter these results.

Recent macaque forelimb experiments (*Yaron et al., 2020*) show that the magnitude of the vectors in the $D(a + b)$ fields is larger than expected from $D(a) + D(b)$ (supralinear summation). We found no evidence for this effect, suggesting that it depends on mechanisms beyond those present in our model.

## Discussion

### Summary of findings and predictions

We have presented a model of the long loop reflex with a main assumption: negative feedback configured with two differential Hebbian learning rules. One novel rule sets the loop's input-output structure, and the other rule (input correlation) promotes stability. We showed that this model can make arm reaches by trying to perceive a given afferent pattern.

Our study made two main points:

1. Many experimental phenomena emerge from a feedback controller with minimally-complete musculoskeletal and neural models (emphasis is placed on the balance between excitation and inhibition).
2. Even if the feedback controller has multiple inputs and outputs, its input-output structure can be flexibly configured by a differential Hebbian learning rule, as long as errors are monotonic.

The first main point above was made using a feedback control network with no learning (called the static network in the Results). We showed that in this static network: (1) reaching trajectories are similar to models of cerebellar ataxia, (2) motor cortex units are tuned to preferred directions, (3) those preferred directions follow a bimodal distribution, (4) motor cortex units present rotational dynamics, (5) reaching is still possible when mass is altered, and (6) spinal stimulation produces convergent direction fields.

The second main point was made using two separate models, both using the same differential Hebbian learning rules, but applied at different locations. The spinal learning model presents the hypothesis that the spinal cord learns to adaptively configure the input-output structure of the feedback controller. The cortical learning model posits that configuring this structure could instead be a function of motor cortex; this would not disrupt our central claims. These two models should not be considered as incompatible hypotheses. Different elements performing overlapping functions are common in biological systems (*Edelman and Gally, 2001*).

Two variations of the spinal learning model in the Appendix show that this learning mechanism is quite flexible, opening the doors for certain types of synergies, and for more complex errors (that still maintain the constraint of monotonicity).

We list some properties of the model, and possible implications:

- Basic arm reaching happens through negative feedback, trying to perceive a target value set in cortex. Learning the input-output structure of the feedback controller may require spinal cord plasticity.

  - Cerebellar patients should not be able to adapt to tasks that require fast reactions, as negative feedback alone cannot compensate for delays in the system (*Sanguineti et al., 2003*). On the other hand, they should be able to learn tasks that require remapping afferent inputs to movements. One example is *Richter et al., 2004*, where cerebellar patients learned to move in a novel dynamic environment, but their movements were less precise than those of controls.

- The shape of reaches is dominated by mechanical and viscoelastic properties of the arm and muscles.

  - Unfamiliar viscous forces as in *Richter et al., 2004* should predictably alter the trajectory (*Figure 3*) for cerebellar patients, who should not be able to adapt unless they move slowly and are explicitly compensating.

- Preferred Directions (PDs) in motor cortex happen because muscles need to contract more when reaching in certain directions.
  - The PD distribution should align with the directions where the muscles need to contract to reduce the error. These directions depend on which error is encoding. If the error is not related to reaching (e.g. related to haptic feedback), a different PD distribution may arise after overtraining.
  - Drift in the PD vectors comes from the ongoing adaptation, and it should not disrupt performance.

- The oscillations intrinsic to delayed feedback control after the onset of a target are sufficient to explain the quasi-oscillations observed in motor cortex (*Churchland et al., 2012*; *Kalidindi et al., 2021*).

- Convergent force fields happen naturally in musculoskeletal systems when there is balance in the stimulation between agonists and antagonists. Linear addition of force fields is a result of forces/torques adding linearly.

Since our relatively simple model reproduces these phenomena, we believe it constitutes a good null hypothesis for them. But beyond explaining experimental observations, this model makes inroads into the hard problem of how the central nervous system (CNS) can generate effective control signals, recently dubbed the 'supraspinal pattern formation' problem (*Bizzi and Ajemian, 2020*). From our perspective, the CNS does not need to generate precise activation patterns for muscles and synergies; it needs to figure out which perceptions need to change. It is subcortical structures that learn the movement details. The key to make such a model work is the differential Hebbian learning framework in *Verduzco-Flores et al., 2022*, which handles the final credit assignment problem.

We chose not to include a model of the cerebellum at this stage. Our model reflects the brain structure of an infant baby who can make clumsy reaching movements. At birth the cerebellum is incomplete and presumably not functional. It requires structured input from spinal cord and cortex to establish correct synaptic connections during postnatal development and will contribute to smooth reaching movements at a later age.

Encompassing function, learning, and experimental phenomena in a single simple model is a promising start towards a more integrated computational neuroscience. We consider that such models have the potential to steer complex large-scale models so they can also achieve learning and functionality from scratch.

## Methods

Simulations were run in the Draculab simulator (*Verduzco-Flores and De Schutter, 2019*). All the parameters from the equations in this paper are presented in the Appendix. Parameters not shown can be obtained from Python dictionaries in the source code. This code can be downloaded from: https://gitlab.com/sergio.verduzco/public_materials/-/tree/master/adaptive_plasticity.

### Unit equations

With the exception of the $A$ and $S_P$ populations, the activity $u_i$ of any unit in *Figure 1* has dynamics:

$$\tau_u \dot{u}_i = \sigma(I) - u_i, \tag{5}$$

$$\sigma(I) = \frac{1}{1 + \exp(\beta(I - \eta))}, \tag{6}$$

where $\tau$ is a time constant, $\beta$ is the slope of the sigmoidal function, $\eta$ is its threshold, and $I = \sum_j \omega_{ij} u_j(t - \Delta t_j)$ is the sum of delayed inputs times their synaptic weights.

Units in the $CE, CI$ populations (in the spinal learning model) or in $M$ (in the cortical learning model) had an additional noise term, which turned *Equation 5* into this Langevin equation:

$$du_i(t) = \frac{1}{\tau_u} \left( \sigma(I) - u_i(t) \right) + \varsigma dW(t), \tag{7}$$

where $W(t)$ is a Wiener process with unit variance, and $\varsigma$ is a parameter to control the noise amplitude. This equation was solved using the Euler-Maruyama method. All other unit equations were integrated using the forward Euler method. The equations for the plant and the muscles were integrated with SciPy's (https://scipy.org/) explicit Runge-Kutta 5(4) method.

Units in the $A$ population use a rectified logarithm activation function, leading to these dynamics for their activity:

$$\tau_a \dot{a} = \log([1 + I - T]_+) - a, \tag{8}$$

where $\tau_a$ is a time constant, $I$ is the scaled sum of inputs, $T$ is a threshold, and $[x]_+ = \max(x, 0)$ is the "positive part" function.

### Learning rules

The learning rule for the connections from $M$ to $CE, CI$ units in the spinal learning model was first described in *Verduzco-Flores et al., 2022*. It has an equation:

$$\dot{\omega}_{ij}(t) = -\Big(\ddot{e}_j(t) - \langle\ddot{e}(t)\rangle\Big)\Big(\dot{c}_i(t - \Delta t) - \langle\dot{c}(t - \Delta t)\rangle\Big). \tag{9}$$

In this equation, $e_j(t)$ represents the activity of the $j$-th unit in $M$ at time $t$, and $\ddot{e}_j(t)$ is its second derivative. Angle brackets denote averages, so that $\langle\ddot{e}\rangle \equiv \frac{1}{N_M}\sum_k \ddot{e}_k$, where $N_M$ is the number of $M$ units. $\dot{c}_i(t)$ is the derivative of the activity for the postsynaptic unit, and $\Delta t$ is a time delay ensuring that the rule captures the proper temporal causality. In the Supplementary Discussion of the Appendix we elaborate on how such a learning rule could be present in the spinal cord.

The learning rule in 9 was also fitted with soft weight-bounding to prevent connections from changing sign, and multiplicative normalization was used to control the magnitude of the weights by ensuring two requirements: (1) all weights from projections of the same $M$ unit should add to $w_{sa}$, (2) all weights ending at the same $C$ unit should add to $w_{sb}$. With this, the learning rule adopted the form:

$$\dot{\omega}_{ij} = -\alpha\omega_{ij}\Big(-\Delta + \lambda\big[(0.5(\zeta_{sa} + \zeta_{sb}) - 1)\big]\Big), \tag{10}$$

In this equation $\alpha$ is a constant learning rate, $\Delta$ is the right-hand side expression of *Equation 9*, and $\lambda$ is a scalar parameter. The value $\zeta_{sa}$ is $w_{sa}$ divided by the sum of outgoing weights from the $j$-th $M$ unit, and $\zeta_{sb}$ is $w_{sb}$ divided by the sum of incoming $M$ weights on $c_i$. This type of normalization is meant to reflect the competition for resources among synapses, both at the presynaptic and postsynaptic level.

The synapses in the connections from $A$ to $M$ and from $A$ to $C$ used the input correlation rule (*Porr and Wörgötter, 2006*):

$$\dot{w} = \alpha_{IC}wI_A\dot{I}_{PA}, \tag{11}$$

where $I_A$ is the scaled sum of inputs from the $A$ population, $\alpha_{IC}$ is the learning rate, $I_{PA}$ is the scaled sum of inputs from $S_{PA}$ or $M$, and $\dot{I}_{PA}$ is its derivative. Unlike the original input correlation rule, this rule uses soft weight bounding to avoid weights changing signs. Moreover, the sum of the weights was kept close to a $\omega_s$ value. In practice this meant dividing the each individual $w$ value by the sum of weights from $A$-to-$M$ (or $A$-to-$C$) connections, and multiplying times $\omega_s$ at each update. In addition, weight clipping was used to keep individual weights below a value $\omega_{max}$.

The learning rule in the cortical learning model was the same, but the presynaptic units were in $S_{PA}$, and the postsynaptic units in $M$.

## Exploratory mechanism

Without any additional mechanisms the model risked getting stuck in a fixed arm position before it could learn. We included two mechanisms to permit exploration in the system. We describe these two mechanisms as they were applied to the spinal learning model and its two variations. The description below also applies to the case of the cortical learning model, with the $M$ units (instead of the $C$ units) receiving the noise and extra connections.

The first exploratory mechanism consists of intrinsic noise in the $CE$ and $CI$ interneurons, which causes low-amplitude oscillations in the arm. We have observed that intrinsic oscillations in the $CE, CI$ units are also effective to allow learning (data not shown), but the option of intrinsic noise permits the use of simple sigmoidal units in $C$, and contributes to the discussion regarding the role of noise in neural computation.

The second mechanism for exploration consists of an additional unit, called $ACT$. This unit acted similarly to a leaky integrator of the total activity in $S_{PA}$, reflecting the total error. If the leaky integral of the $S_{PA}$ activity crossed a threshold, then $ACT$ would send a signal to all the $CE$ and $CI$ units, causing adaptation. The adaptation consisted of an inhibitory current that grew depending on the accumulated previous activity.

To model this, $CE$ and $CI$ units received an extra input $I_{adapt}$. When the input from the $ACT$ unit was larger than 0.8, and $I_{adapt} < 0.2$, the value of $I_{adapt}$ would be set to $(u_i^{slow})^2$. This is the square of a low-passed filtered version of $u_i$. More explicitly,

$$\tau_{slow}\dot{u}_i^{slow} = u_i - u_i^{slow}. \tag{12}$$

If the input from $ACT$ was smaller than 0.8, or $I_{adapt}$ became larger than 0.2, then $I_{adapt}$ would decay towards zero:

$$\tau_{slow}\dot{I}_{adapt} = -I_{adapt}.$$  (13)

With this mechanism, if the arm got stuck then error would accumulate, leading to adaptation in the spinal interneurons. This would cause the most active interneurons to receive the most inhibition, shifting the 'dominant' activities, and producing larger amplitude exploratory oscillations.

When a new target is presented, $ACT$ must reset its own activity back to a low value. Given our requirement to fully implement the controller using neural elements, we needed a way to detect changes in $S_P$. A unit denominated $CHG$ can detect these changes using synapses that react to the derivative of the activity in $S_P$ units. $CHG$ was connected to $ACT$ in order to reset its activity.

More precisely, when inputs from $CHG$ were larger than 0.1, the activity of $ACT$ had dynamics:

$$\dot{a}(t) = -40a(t).$$  (14)

Otherwise it had these dynamics:

$$\dot{a}(t) = a(t)\big(\sigma(I) - \theta_{ACT}\big), \text{ if } \sigma(I) < \theta_{ACT},$$  (15)

$$\tau_{ACT}\dot{a}(t) = \big(\sigma(I) - \theta_{ACT}\big)\big[1 - a(t) + \gamma\dot{\sigma}(I)\big], \text{ otherwise.}$$  (16)

As before, $\sigma(\cdot)$ is a sigmoidal function, and $I$ is the scaled sum of inputs other than $CHG$. When $\sigma(I)$ is smaller than a threshold $\theta_{ACT}$ the value of $a$ actually decreases, as this error is deemed small enough. When $\sigma(I) > \theta_{ACT}$ the activity increases, but the rate of increase is modulated by a rate of increase $\dot{\sigma}(I) \equiv \sigma(I) - \sigma(\tilde{I})$, where $\tilde{I}$ is a low-pass filtered version of $I$ is a constant parameter.

$CHG$ was a standard sigmoidal unit receiving inputs from $S_P$, with each synaptic weight obeying this equation:

$$\omega_j(t) = \alpha|\dot{s}_j(t)| - \omega_j(t),$$  (17)

where $s_j$ represents the synapse's presynaptic input.

## Plant, muscles, afferents

The planar arm was modeled as a compound double pendulum, where both the arm and forearm were cylinders with 1 kg. of mass. No gravity was present, and a moderate amount of viscous friction was added at each joint ($3 \frac{N\,m\,s}{rad}$). The derivation and validation of the double pendulum's equations can be consulted in a Jupyter notebook included with Draculab's source code (in the tests folder).

The muscles used a standard Hill-type model, as described in *Shadmehr and Wise, 2005*, Pg. 99. The muscle's tension $T$ obeys:

$$\dot{T} = \frac{K_{SE}}{b}\left[g \cdot I + K_{PE}\Delta x + b\dot{x} - \left(1 + \frac{K_{PE}}{K_{SE}}\right)T\right],$$  (18)

where $I$ is the input, $g$ an input gain, $K_{PE}$ the parallel elasticity constant, $K_{SE}$ the series elasticity constant, $b$ is the damping constant for the parallel element, $x$ is the length of the muscle, and $\Delta x = x - x_1^* - x_2^*$. In here, $x_1^*$ is the resting length of the series element, whereas $x_2^*$ is the resting length of the parallel element. All resting lengths were calculated from the steady state when the hand was located at coordinates (0.3, 0.3).

We created a model of the Ia and II afferents using simple structural elements. This model includes, for each muscle one dynamic nuclear bag fiber, and one static bag fiber. Both of these fibers use the same tension equation as the muscle, but with different parameters. For the static bag fiber:

$$\dot{T}^s = \frac{K_{SE}^s}{b^s}\left[K_{PE}^s\Delta x + b^s\dot{x} - \left(1 + \frac{K_{PE}^s}{K_{SE}^s}\right)T^s\right].$$  (19)

The dynamic bag fiber uses the same equation, with the $s$ superscript replaced by $d$. No inputs were applied to the static or dynamic bag fibers, so they were removed from these equations. The rest lengths of the static and dynamic bag fibers where those of their corresponding muscles times factors $l_0^s, l_0^d$, respectively.

The Ia afferent output is proportional to a linear combination of the lengths for the serial elements in both dynamic and static bag fibers. The II output has two components, one proportional to the

length of the serial element, and one approximately proportional to the length of the parallel element, both in the static bag fiber. In practice this was implemented through the following equations:

$$I_a = g_{I_a} \left[ \left( \frac{f_s^{I_a}}{K_{SE}^s} \right) T^s + \left( \frac{1 - f_s^{I_a}}{K_{SE}^d} \right) T^d \right], \tag{20}$$

$$II = g_{II} \left[ \left( \frac{f_s^{II}}{K_{SE}^s} \right) T^s + \left( \frac{1 - f_s^{II}}{K_{PE}^s} \right) \left( T^s - b^s \dot{x} \right) \right]. \tag{21}$$

In here, $g_{I_a}$ and $g_{II}$ are gain factors. $f_s^{I_a}$ and $f_s^{II}$ are constants determining the fraction of $I_a$ and $II$ output that comes from the serial element.

The model of the Golgi tendon organ producing the Ib outputs was taken from **Lin and Crago, 2002**. First, a rectified tension was obtained as:

$$r = g_{I_b} \log(T^+ / T_0 + 1). \tag{22}$$

$g_{I_b}$ is a gain factor, $T_0$ is a constant that can further alter the slope of the tension, and $T^+ = \max(T, 0)$ is the tension, half-rectified. The $I_b$ afferent output followed dynamics:

$$\tau_{I_b} \dot{I_b} = r - I_b. \tag{23}$$

## Static connections

In all cases, the connections to $S_A$ used one-to-one connectivity with the $A$ units driven by the II afferents, whereas connections from $A$ to $M$ and $C$ used all-to-all projections from the units driven by the Ia and Ib afferents. Projections from $S_A$ to $S_{PA}$ used one-to-one excitatory connections to the first 6 units, and inhibitory projections to the next six units. Projections from $S_P$ to $S_{PA}$ used the opposite sign from this.

Connections from $S_{PA}$ to $M$ were one-to-one, so the $j$-th unit in $S_{PA}$ only sent a projection to unit $j$ in $M$. A variation of this connectivity is presented in the Appendix (See *Variations of the spinal learning model*).

We now explain how we adjusted the synaptic weights of the static network. To understand the projections from $M$ to $C$ and to the alpha motoneurons it is useful to remember that each $CE, CI, \alpha$ trio is associated with one muscle, and the $M$ units also control the error of a single muscle. This error indicates that the muscle is longer than desired. Thus, the $M$ unit associated with muscle $i$ sent excitatory projections to the $CE$ and $\alpha$ units associated with muscle $i$, and to the $CI$ units of the antagonists of $i$. Additionally, weaker projections were sent to the $CE, \alpha$ units of muscle $i$'s agonists. Notice that only excitatory connections were used.

The reverse logic was used to set the connections from $A$ to $C$ and $M$. If muscle $i$ is tensing or elongating, this can predict an increase in the error for its antagonists, which is the kind of signal that the input correlation rule is meant to detect. Therefore, the $Ib$ afferent (signaling tension) of muscle $i$ sent an excitatory signal to the $CI$ unit associated with muscle $i$, and to the $CE, \alpha$ units associated with $i$'s antagonists. Moreover, this $Ib$ afferent also sent an excitatory projection to the dual of the $M$ unit associated with muscle $i$. Connections from $Ia$ afferents (roughly signaling elongation speed) followed the same pattern, but with slightly smaller connection strengths.

## Rotational dynamics

We explain the method to project the activity of $M$ onto the jPCA plane. For all units in $M$ we considered the activity during a 0.5 s sample beginning 50 ms after the target onset. Unlike (**Churchland et al., 2012**), we did not apply PCA preprocessing, since we only have 12 units in $M$. Let $m_{i,j,k,t}$ be the activity at time $t$ of the unit $i$ in $M$, when reaching at target $j$ for the $k$-th repetition. By $m_{i,j,\langle k \rangle,t}$ we denote the average over all repeated reaches to the same target, and by $m_{i,\langle j \rangle,\langle k \rangle,t}$ we indicate averaging over both targets and repetitions. The normalized average trace per condition is defined as: $m_{i,j}(t) \equiv m_{i,j,\langle k \rangle,t} - m_{i,\langle j \rangle,\langle k \rangle,t}$. Let $I$ stand for the number of units in $M$, $T$ for the number of time points, and $J$ for the number of targets. Following (**Churchland et al., 2012**), we unroll the set of $m_{i,j}(t)$ values into a matrix $X \in R^{JT \times I}$, so we may represent the data through a matrix $M$ that provides the least-squares solution to the problem $\dot{X} = XM$. This solution comes from the equation $\hat{M} = (X^T X)^{-1} X^T \dot{X}$. Furthermore, this matrix can be decomposed into symmetric and anti-symmetric

components $M_{symm} = (\hat{M} + \hat{M}^T)/2, M_{skew} = (\hat{M} - \hat{M}^T)/2$. The jPCA plane comes from the complex conjugate eigenvalues of $M_{skew}$.

In practice, our source code follows the detailed explanation provided in the Supplementary Information of *Churchland et al., 2012*, which reformulates this matrix problem as a vector problem.

## Parameter search

We kept all parameter values in a range where they still made biological sense. Parameter values that were not constrained by biological data were adjusted using a genetic algorithm, and particle swarm optimization (PSO). We used a separate optimization run for each one of the configurations, consisting of roughly 30 iterations of the genetic and PSO algorithms, with populations sizes of 90 and 45 individuals respectively. After this we manually adjusted the gain of the control loop by increasing or decreasing the slope of the sigmoidal units in the $M$ and $S_A$ populations. This is further described in the Appendix (*Gain and oscillations* section).

The parameters used can affect the results in the paper. We chose parameters that minimized either the error during the second half of the learning phase, or the error during center-out reaching. Both of these measures are agnostic to the other results.

## Preferred direction vectors

Next we describe how PD vectors were obtained for the $M$ units.

Let $m_{jk}$ denote the firing rate of the $j$-th $M$ unit when reaching for the $k$-th target, averaged over 4 s, and across reaches to the same target. We created a function $h_j : R^2 \rightarrow R$ that mapped the X,Y coordinates of each target to its corresponding $m_{jk}$ value, but in the domain of $h_j$ the coordinates were shifted so the center location was at the origin.

Next we approximated $h_j$ with a plane, using the least squares method, and obtained a unit vector $u_j$ normal to that plane, starting at the intersection of the $z$-axis and the plane, and pointing towards the XY plane. The PD vector was defined as the projection of $u_j$ on the XY plane.

In order to predict the PD vectors, we first obtained for each muscle the 'direction of maximum contraction', verbally described in panel B of *Figure 5*. More formally, let $l_{ik}$ denote the length of the $i$-th muscle when the hand is at target $k$, and let $l_i^0$ denote its length when the hand is at the center location. With $\bar{r}_k$ we denote the unit vector with base at the center location, pointing in the direction of the $k$-th target. The direction of maximum length change for the $i$-th muscle comes from the following vector sum:

$$\bar{v}_i = \sum_{k=1}^{8} \left[ \frac{l_i^0 - l_{ik}}{l_i^0} \right]_+ \bar{r}_k, \tag{24}$$

where $[x]_+ = \max(x, 0)$.

For the $j$-th unit in $M$, its predicted PD vector comes from a linear combination of the $\bar{v}_i$ vectors. Let the input to this unit be $\sum_i w_{ji} e_i$, where $e_i$ is the output of the $i$-th SPF unit (representing the error in the $i$-th muscle). The predicted PD vector is:

$$\bar{d}_j = \sum_{i=0}^{5} w_{ji} \bar{v}_i \tag{25}$$

To obtain the main axis of the PD distribution, the $i$-th PD vector was obtained in the polar form $(r_i, \theta_i)$, with $\theta \in [-\pi, \pi]$. We reflected vectors in the lower half using the rule: $\theta_i^* = \theta_i + \pi$ if $\theta_i < 0$, $\theta_i^* = \theta_i$ otherwise. The angle of the main axis was the angle of the average PD vector using these modified angles: $\theta_{main} = \arctan\left(\frac{\sum_i r_i \sin\theta_i^*}{\sum_i r_i \cos\theta_i^*}\right)$.

## Statistical tests

To find whether $M$ units were significantly tuned to the reach direction we used a bootstrap procedure. For each unit we obtained the length of its PD vector 10,000 times when the identity of the target for each reach was randomly shuffled. We considered there was significant tuning when the length of the true PD vector was longer than 99.9% of these random samples.

To obtain the coefficient of determination for the predicted PD angles, let $\theta^j_{true}$ denote the angle of the true PD for the $j$-th $M$ unit, and $\theta^j_{pred}$ be the angle of its predicted PD. We obtained residuals for the angles as $\epsilon_j = \theta^j_{true} - \theta^j_{pred}$, where this difference is actually the angle of the smallest rotation that turns one angle into the other. Each residual was then scaled by the norm of its corresponding PD vector, to account for the fact that these were not homogeneous. Denoting these scaled residuals as $\epsilon^*_j$ the residual sum of squares is $SS_{res} = \sum_j (\epsilon^*_j)^2$. The total sum of squares was: $SS_{tot} = \sum_j (\theta^j_{true} - \bar{\theta}_{true})^2$, where $\bar{\theta}_{true}$ is the mean of the $\theta^j_{true}$ angles. The coefficient of determination comes from the usual formula $R^2 = 1 - \frac{SS_{res}}{SS_{tot}}$.

To assess bimodality of the PD distribution we used a version of the Rayleigh statistic adapted to look for bimodal distributions where the two modes are oriented at 180 degrees from each other, introduced in *Lillicrap and Scott, 2013*. This test consists of finding an modified Rayleigh $r$ statistic defined as:

$$r = \frac{1}{N}\left( \left(\sum_{i=1}^{N} cos(2\phi_i)\right)^2 + \left(\sum_{i=1}^{N} cos(2\phi_i)\right)^2 \right), \tag{26}$$

where the $\phi_i$ angles are the angles for the PDs. A bootstrap procedure is then used, where this $r$ statistic is produced 100,000 times by sampling from the uniform distribution on the $(0, \pi)$ interval. The PD distribution was deemed significantly bimodal if its $r$ value was larger than 99.9% of the random $r$ values.

We used a bootstrap test to find whether there was statistical significance to the linear addition of direction fields. To make this independent of the individual pair of locations stimulated, we obtained the direction fields for all 15 possible pairs of locations, and for each pair calculated the mean angle difference between $D(a+b)$ and $D(a) + D(b)$ as described in the main text. We next obtained the mean of these 15 average angle deviations, to obtain a global average angle deviation $\gamma_{global}$.

We then repeated this procedure 400 times when the identities of the stimulation sites $a, b$ were shuffled, to obtain 400 global average angle deviations $\gamma^j_{global}$. We declared statistical significance if $\gamma_{global}$ was smaller than 99% of the $\gamma^j_{global}$ values.

## Acknowledgements

The authors wish to thank Prof. Kenji Doya for helping revise early versions of this manuscript.

## Additional information

### Funding
No external funding was received for this work.

### Author contributions
Sergio Oscar Verduzco-Flores, Conceptualization, Software, Formal analysis, Validation, Investigation, Visualization, Writing - original draft, Writing - review and editing; Erik De Schutter, Conceptualization, Resources, Supervision, Methodology, Project administration, Writing - review and editing

### Author ORCIDs
Sergio Oscar Verduzco-Flores (iD) http://orcid.org/0000-0002-0712-145X
Erik De Schutter (iD) http://orcid.org/0000-0001-8618-5138

### Decision letter and Author response
Decision letter https://doi.org/10.7554/eLife.77216.sa1
Author response https://doi.org/10.7554/eLife.77216.sa2

## Additional files

### Supplementary files
• Transparent reporting form

### Data availability
The current manuscript is a computational study, so no data have been generated for this manuscript. The source code to generate all figures is available as two commented Jupyter notebooks. They can be downloaded from the following repository: https://gitlab.com/sergio.verduzco/public_materials/-/tree/master/adaptive_plasticity (copy archived at swh:1:rev:482c0659d6e90b30a4a1ac-b4ab3e3a03dfd902c4). Instructions are in the "readme.md" file. Briefly: Prerequisites for running the notebooks are: Python 3.5 or above (https://www.python.org); Jupyter (https://jupyter.org); Draculab (https://gitlab.com/sergio.verduzco/draculab). Please see the links above for detailed installation instructions.

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

## Appendix 1

## Supplementary discussion

Comparison with previous models

There are many other models of reaching and motor control. Most of them have one or more of the following limitations:

1. They use non-neural systems to produce motor commands.
2. They control a single degree of freedom, sidestepping the problem of controller configuration, since the error is one-dimensional.
3. They do not model a biologically plausible form of synaptic learning.

We will contrast our model with some of the work that does not strongly present these limitations, and with a few others. Due to space constraints the contributions of many models will not be addressed, and for those mentioned we will limit ourselves to explain some of their limitations.

The model in *DeWolf et al., 2016* is similar has similar goals to our model, but with very different assumptions. In their model, motor cortex receives a location error vector $x$, and transforms it into a vector of joint torques. So that this transformation implements adaptive kinematics, it must approximate a Jacobian matrix that includes the effects of the arm's inertia matrix, using location errors and joint velocities as training signals. This is accomplished by adapting an algorithm taken from the robotics literature (*Cheah et al., 2006*), implementing it in a spiking neural network. Additionally a second algorithm from robotics (*Sanner and Slotine, 1992*) is used to provide an adaptive dynamics component, which is interpreted as the cerebellar contributions.

In order to implement vector functions in spiking neural networks, *DeWolf et al., 2016* uses the Neural Engineering Framework (*Bekolay et al., 2014*). The essence of this approach is to represent values in populations of neurons with cosine-like tuning functions. These populations implement expansive recoding, becoming a massively overcomplete basis of the input space. Implementing a function using this population as the input is akin to using a linear decoder to extract the desired function values from the population activity. This can be done through standard methods, such as least-squares minimization, or random gradient descent. The parameters of the linear decoder then become weights of a feedforward neural layer implementing the function.

The model in *DeWolf et al., 2016* has therefore a rather different approach. They use engineering techniques to create a powerful motor control system, using algorithms from robotics, and 30,000 neurons to approximate their computations, which are then ascribed to sensory, motor, and premotor cortices, as well as the cerebellum. In contrast, we use 74 firing rate units, and unlike (*DeWolf et al., 2016*) we include muscles, muscle afferents, transmission delays, and a spinal cord.

There is nothing intrinsically wrong with using an engineering approach to try to understand a biological function. The crucial part is which model will be experimentally validated. Some differences between the models that may be able to separate them experimentally are: (1) In *DeWolf et al., 2016* premotor cortex is required to produce the error signal, whereas we ascribed this to sensory cortex. (2) In *DeWolf et al., 2016* direct afferent connections to motor cortex are not considered, whereas in our model they are important to maintain stability during learning (in the absence of a cerebellum). (3) In *DeWolf et al., 2016* spinal cord adaptation is not necessary to implement adaptive kinematics. In contrast, spinal cord adaptation is important in one of the interpretations of our model.

The model in *Dura-Bernal et al., 2015* uses spiking neurons, and a realistic neuromechanical model in order to perform 2D reaching. The feedback is in term of muscle lengths, rather than muscle afferent signals. There is no mechanism to stop the arm, or hold it on target. Most importantly, learning relies on a critic, sending rewarding or punishing signals depending on whether the hand was approaching or getting away from the target. This is implicitly reducing the error dimension using a hidden mechanism. Furthermore, each single target must be trained individually, and it is not discussed how this can lead to a flexible reaching mechanism without suffering from catastrophic interference.

The model in *Todorov, 2000* is used to obtain characteristics of M1 activity given the required muscle forces to produce a movement. It is an open-loop, locally-linear model, where all connections from M1 directly stimulate a linear motoneuron. Among other things, it showed that representations

of kinematic parameters can appear when the viscoelastic properties of the muscles are taken into account, giving credence to the hypothesis that M1 directly activates muscle groups. Outside of its scope are neural implementation, learning, or the role of spinal cord.

*Li et al., 2005* proposes a 2-level hierarchical controller for a 2-DOF arm. Since this model is based on optimal control theory, it is given a cost function, and proceeds by iteratively approaching a solution of the associated Jacobi-Bellman equation. There is no neural implementation of these computations, nor a description of learning.

*Martin et al., 2009* is not properly a neural model, but it is rooted in Dynamic Field Theory, which assumes that the population of neuronal activities encodes "activation variables". For this model activation variables represent kinematic parameters as a virtual joint location vector $\lambda(t)$, which is an internal representation of the arm's configuration.

The innovative part of *Martin et al., 2009* is in describing how $\lambda(t)$ is updated and used to control a redundant manipulator. In particular, the kinematic Jacobian, its null-space matrix, their derivatives, and the Moore-Penrose pseudoinverse are all computed analytically in order to obtain differential equations where the joint motions that move the end effector decouple from those which don't.

Encapsulating the muscle commands into a virtual joint location whose dynamics are decoupled for motions that don't affect the end-effector location is a very interesting concept. Still, this is far from a neural implementation, and learning is not considered.

The model in *Caligiore et al., 2014* studies the long-term development of infant reaching using a PD controller, and an actor critic mechanism implementing a neural version of TD-learning (*Sutton and Barto, 2018*). The 2-dimensional PD controller receives two desired joint angles (interpreted as an equilibrium position), producing appropriate torques. Since it uses the Equilibrium Point (EP) Hypothesis (*Feldman, 1986*), the reaching portion of this model is tantamount to associating states with equilibrium positions. This model thus performs at a higher level of analysis. Our model could operate at the same level if we added a reinforcement learning component to learn $S_P$ values allowing the hand to *touch* a target whose distance is known. (*Caligiore et al., 2014*) does not consider separate neuronal populations (e.g. spinal cord, sensory cortex), propagation delays, or low-level learning.

The model in *Izawa et al., 2004* shows how reinforcement learning can be applied to redundant actuators, such as biological arms. It is, however, not a neural model.

In *Mici et al., 2018*, a neural network produces predictions of visual input in order to deal with temporal delays in a sensorimotor system. The network used for this study uses non-local learning, and adds or destroys nodes as required during its operation. It is thus not biologically-plausible.

*Tsianos et al., 2014* is a reaching model that also considers the spinal cord as a highly-configurable controller. Corticospinal inputs are assumed to be step commands, which means that motor cortex operates in an open-loop configuration. In order to produce reaching based on these constant commands, a biologically-implausible gradient descent mechanism is required, where the same reach is performed for various values of a synaptic weight, keeping the value that led to the best performance. Furthermore, the model learns to reach one target at a time, which would require learning anew when the new target is more than 45 degrees apart.

As mentioned in the main text, in the context of rotational dynamics, the model in *Sussillo et al., 2015* was used to produce desired muscle forces using a recurrent neural network. This model uses the FORCE algorithm (*Sussillo and Abbott, 2009*) to adjust the weights of a neural network with activity vector $\mathbf{r}(t)$ so it can produce experimentally observed muscle activity $\mathbf{m}(t)$. Oscillatory dynamics arise when the model is constrained to be simple.

Although very insightful, this model is limited by the fact that *Equations 2 and 3* represent an open-loop configuration, where only the M1 dynamics are considered. In essence, the model is doing a function approximation with the FORCE algorithm. The question of how the training data $\mathbf{m}(t)$ is produced is not addressed, nor is the role of spinal cord or sensory cortex (but see their Supplementary Figure 1).

Other than the aforementioned model in *Tsianos et al., 2014*, we are unaware of spinal cord models addressing arm reaching. When these models are coupled with a musculoskeletal system, it is usually for the control of one degree of freedom using antagonistic neural populations. We mention some examples next.

The spinal cord model in *Bashor, 1998* inspired much of the subsequent work, organizing the spinal circuit into six pairs of populations controlling two antagonistic muscle groups at one joint. With this model, the effect of Ia and Ib afferent input was studied in various neuronal populations.

*Stienen et al., 2007* used a model similar to that in *Bashor, 1998* in conjunction with a one DOF musculoskeletal model to study the effect of Ia afferents in the modulation of reflexes. *Cutsuridis, 2007* also adapted a similar model, and used it to inquire whether Parkinsonian rigidity arose from alterations in reciprocal inhibition mediated by reduced dopamine.

The model in *Cisi and Kohn, 2008* has several features not found in *Bashor, 1998*, including a heterogeneous motoneuron population, and a mechanism to generate electromyograms. This was used to study the generation of the H-reflex, and response properties of the motor neuron pool (*Farina et al., 2014*).

The model in *Shevtsova et al., 2015* goes beyond models such as *Bashor, 1998* by using populations of genetically-identified interneurons. This model is used to study rhythm generating abilities of the spinal circuit, as well as coordination between flexors and extensors (*Shevtsova and Rybak, 2016*; *Danner et al., 2017*). Knowledge regarding the role of genetically identified spinal interneurons in movement generation is still evolving (*Zelenin et al., 2021*; *Stachowski and Dougherty, 2021*, e.g.).

This paper focuses on mammals, but the spinal cord of simpler organisms is better characterized (*Borisyuk et al., 2011*; *Cangiano and Grillner, 2005*, e.g.), and may lead to the first realistic models producing ethological behavior.

## Possible implementations of the learning rule

The learning rule in *Equation 9* is a Hebbian rule that also presents derivatives, heterosynaptic competition, and normalization (e.g. removing the mean) of its terms. None of these is new in a model claiming biological plausibility (*Porr and Wörgötter, 2006*; *Zappacosta et al., 2018*; *Fiete et al., 2010*; *Kaleb et al., 2021*, e.g.). We nevertheless mention possible ways for derivatives and normalized terms to appear.

Formally, the time derivative of a function $f : R \to R$ evaluated at time $t$ is the limit $\frac{f(t+\Delta t)-f(t)}{\Delta t}$ as $\Delta t \to 0$. If $f$ represents the firing rate of a cell, a measure of change roughly proportional to the derivative can come from $f(t) - f(t - \Delta t)$ for some small value $\Delta t$. The most obvious way that such a difference may arise is through feedforward inhibition (for the $e_j$ signal), and feedback inhibition (for the $c_i$ signal). Feedforward and feedback inhibition are common motifs in spinal circuits (*Pierrot-Deseilligny and Burke, 2005*).

A somewhat different way to approach a time derivative is by using two low-pass filters with different time constants:

$$\frac{df}{dt} \approx f_{fast}(t) - f_{slow}(t),$$

where

$$\tau_1 \dot{f}_{fast}(t) = f(t) - f_{fast}(t), \quad \tau_2 \dot{f}_{slow}(t) = f(t) - f_{slow}(t), \quad \tau_2 \gg \tau_1.$$

These principles are illustrated in *Lim and Goldman, 2013*, where they are used to explain negative-derivative feedback.

Low-pass filtering can also arise in the biochemical cascades following synaptic depolarization. The most salient case is intracellular calcium concentration, which has been described as an indicator of firing rate with leaky integrator dynamics (*Helmchen, 1999*). Although the physiology of spinal interneurons has not been characterized with sufficient detail to make specific hypotheses, it is clearly possible that feedback inhibition and low-pass filtering are enough to approximate a second-order derivative.

The $e_j, c_i$ terms in our learning rule are mean-centered. The most straightforward way to subtract a mean is to have inhibitory units with converging inputs (e.g. receiving all the $e_j$ signals) providing input to the $c_i$ units. The Ib interneurons (*Pierrot-Deseilligny and Burke, 2005*) are one possibility for mediating this. Another possibility is that the mean-subtraction happens at the single unit level when the input ($e_j$) and lateral ($c_i$) connections are located at different parts of the dendritic tree. In particular, a larger level of overall input activation $\langle \ddot{e} \rangle$ could produce a scarcity of postsynaptic

resources flowing from the main branches of the dendritic tree into the individual dendritic spines, resulting in reduced plasticity.

## Limitations of the model

A model as simple as ours will undoubtedly be wrong in many details. The main simplifying assumptions of our model are:

- Firing rate encoding. Each unit in this model captures the mean-field activity of a neuronal population. This may occlude computations depending on spike timing, as well as fine-grained computations at the neuronal and dendritic level.
- Trivial sensory cortex. We assumed that sensory cortex conveyed errors directly based on static gamma afferent activity. Sensory cortex may instead build complex representations of afferent activity. It would be interesting to test how the requirement of controllability could guide learning of these representations.
- No visual information. Should a visual error be available, it could be treated by $M$ in a similar way to somatosensory errors. If the visual error holds a monotonic relation with the afferent signals, then it should be possible to adjust the long-loop reflex in order to reduce it. When the relation between the visual error and the afferent signals is not monotonic (e.g. in some context the afferent signals correlate positively, and in other negatively), an alternative approach involving reinforcement learning can be pursued (*Verduzco-Flores et al., 2022*).
- Very gross anatomical detail. The detailed anatomical organization of cortex and spinal cord is not considered. Moreover, the proportions for different types of cells are not considered.
- Errors must be monotonic. Muscle activation may not monotonically reduce visual errors. Moreover, the haptic perception of contact is dependent on the environment, so it would not make an appropriate error signal.
- No cerebellum, basal ganglia, brainstem, or premotor cortex.

Each of these omissions is also a possible route for improving the model. We aim to grow a more sophisticated model, but each new component must integrate with the rest, improve functionality and agree with biological literature.

A final limitation concerns proofs of convergence. Many factors complicate them for this model: transmission delays, noise, synaptic learning, fully neural implementation, as well as complex muscle and afferent models. We tested our results for many initial conditions, but this of course is no guarantee.

50 years ago Marr's model of the cerebellum became a stepping stone to further theoretical and experimental progress, despite all its shortcomings (*Kawato et al., 2021*). We aspire our model to be the next step towards a complete model of motor control.

## Variations of the spinal learning model

The main text mentions two variations of the spinal learning network that emphasize the robustness and potential of the learning mechanism. We will explain the rationale behind those two variations.

There is evidence for interneurons that drive a set of muscles, possibly setting the circuit foundation for motor synergies (*Giszter, 2015*; *Levine et al., 2014*; *Bizzi and Cheung, 2013*). To explore whether our ideas were compatible with interneurons activating multiple muscles, we explored whether reaching can be learned when the $CE$ and $CI$ units send projections to more than one motoneuron. To achieve this we modified the architecture of *Figure 1* so that for every combination of two different muscles there was a pair of $CE, CI$ units that stimulated both of them.

As illustrated in the *Appendix 1—figure 1*, from the set of 6 muscles there are 15 combinations of 2 muscles, but 3 of them consist of antagonist pairs. Removing these we are left with 12 pairs of muscles, and for each muscle pair we had a Wilson-Cowan-like $CE, CI$ pair sending projections to the alpha motoneurons of both muscles. Furthermore, for each pair of muscles, there is another pair that contains both their antagonists, and we can use this fact to generalize the concept of antagonists when interneurons project to several motoneurons. The $CE$ units sent projections to the $CI$ units of their antagonists. This organization allowed us to maintain the balance between excitation and inhibition in the network, along with the connectivity motifs used previously.

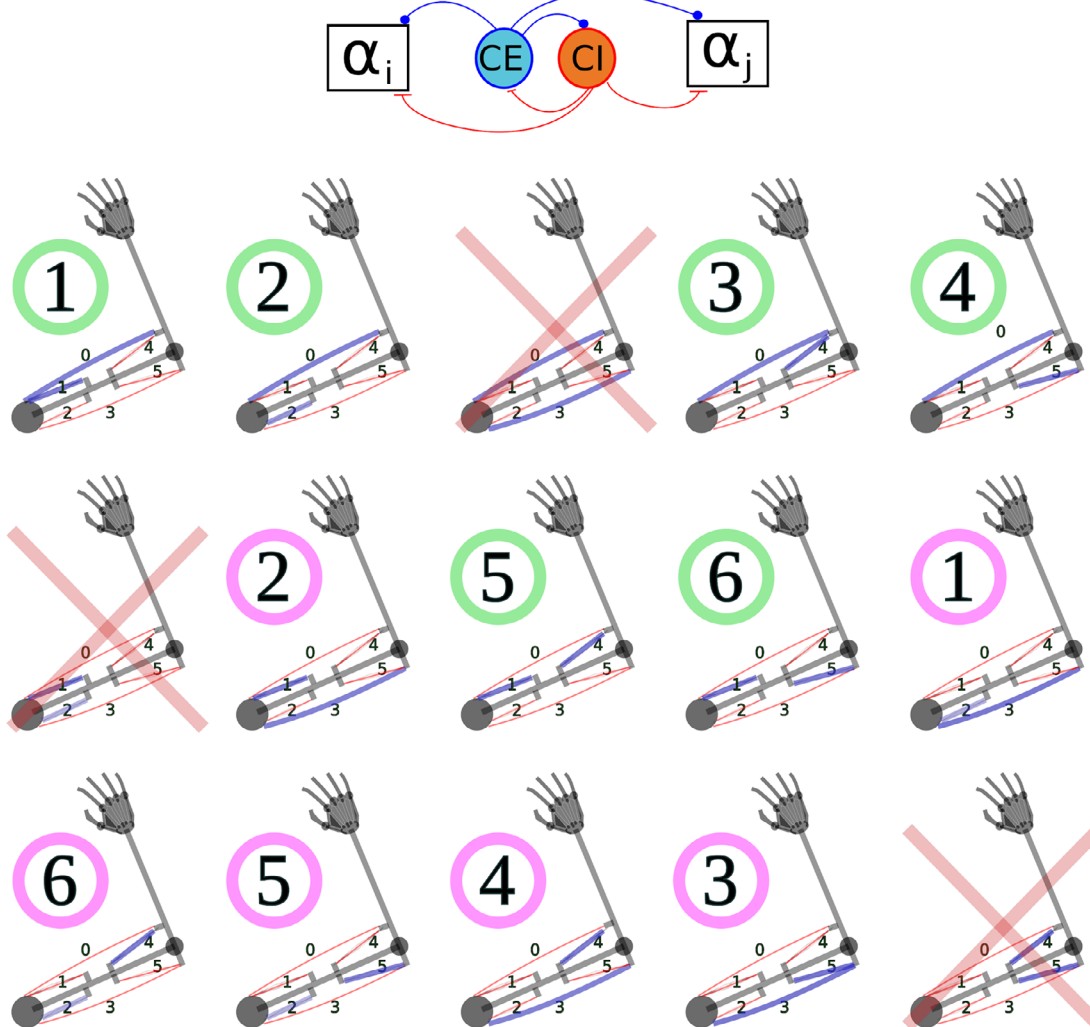

**Appendix 1—figure 1.** Modified architecture of the $C$ population. Each pair of $CE, CI$ units projects to two $\alpha$ motoneurons (top). There are 15 possible pairs of muscles, corresponding to the blue lines for each arm in the figure. Three of the pairs (marked with red crosses) contain antagonist muscles, and are not included. The remaining 12 pairs can be arranged into 2 groups of 6 units each. The units in the group marked with green circles are the antagonists of the units with the same number, marked with pink circles.

Because this model could be considered a proof-of-concept for the compatibility of our learning mechanisms with this particular type of synergies, we refer to this model as the "synergistic" network.

To introduce the second variation of the spinal learning network, we may notice that in all configurations so far the projections from $S_{PA}$ to $M$ use one-to-one connectivity (each $M$ unit controls the length error of one muscle). Interestingly, this is not necessary. In a second variation of the spinal learning network, dubbed the "mixed errors" network, each unit in $M$ can be driven by a linear combination of $S_{PA}$ errors.

To ensure that the information about the $S_{PA}$ activity was transmitted to $M$, we based our $S_{PA}$ to $M$ connections on the following 6-dimensional orthogonal matrix:

$$R = \begin{bmatrix} 1 & 1 & 1 & 1 & 1 & 1 \\ 1 & 1 & 1 & -1 & -1 & -1 \\ 1 & -2 & 1 & -1 & 2 & -1 \\ -1 & -1 & 2 & 2 & -1 & -1 \\ -1 & 1 & 0 & 0 & 1 & -1 \\ -1 & 0 & 1 & -1 & 0 & 1 \end{bmatrix}. \tag{A1}$$

The rows of this matrix form an orthogonal basis in $R^6$, and normalizing each row we obtain a matrix $R^*$ whose rows form an orthonormal basis. Connections from the 12 $S_{PA}$ units to the 12 $M$ units used this 12 × 12 matrix:

$$\begin{bmatrix} R^* & -R^* \\ -R^* & R^* \end{bmatrix}. \tag{A2}$$

The one-to-one connections from $S_{PA}$ to $M$ used in our models are unrealistic, but the mixed errors network shows that this simplification can be overcome, since all the results of the paper also apply to this variation.

All numerical tests applied to the 3 configurations in the main text of the paper were also applied to the two variations of the spinal learning model. Results can be seen in this Appendix, in the Comparison of the 5 configurations. *Figure 2—figure supplements 3 and 4* illustrate the training phase for the synergistic and mixed error networks.

We also tested whether the stimulation of an isolated spinal cord produced convergent direction fields in the synergistic network, as was done in the main text for the spinal network $C$ common to the other four configurations. We found that the mean angle difference $\gamma_{a,b}$ between the direction fields $D(a + b)$ and $D(a) + D(b)$, averaged over the 144 possible $(a, b)$ pairs, was 19.8 degrees.

Randomly choosing pairs $(a, b)$ and $(c, d)$ for the stimulation locations lead to a $\gamma_{a,b}$ angle of 72.3 degrees. As before, a bootstrap test showed that this $\gamma_{a,b}$ value is significantly different ($p < 0.0001$). Removing the resting field does not alter this result. Moreover, we found no evidence for supralinear summation of force fields in the synergistic network.

## The model fails when elements are removed

Due to the larger number of tests, we only used 5 trials for each configuration in this section. The p values reported in this section come from the one-sided t-test. For brevity, the different configurations of the model will be denoted by numbers in the rest of this Appendix: 1=spinal learning, 2=cortical learning, 3=static network, 4=synergistic network, 5=mixed errors network.

A model with fully random connectivity and no plasticity has an exceedingly low probability of having an input-output structure leading it to reduce errors. The configurations of the model with plasticity (configurations 1,2,4,5), however, only have random connections at one of the projections in the sensorimotor loop (either from $M$ to $C$, or from $S_{PA}$ to $M$). This may increase the chance of randomly obtaining a good input-output structure, which could throw the usefulness of the learning rules into question.

Removing both types of plasticity in configurations 1,2,4,5 impaired reaching in all 5 tests for each configuration, as reflected by the inability to reduce the average error below 10 centimeters in any of the last 4 reaches of the learning phase. This was also true when removing only the plasticity in the connections from $M$ to $C$ (configurations 1, 4, 5) or from $S_{PA}$ to $M$ (configuration 2). In each one of the plastic configurations (1,2,4,5) the average error for the last 4 training targets was ($22 \pm 10|22 \pm 4|24 \pm 9|22 \pm 5$) centimeters, which was significantly higher than the case with normal plasticity ($p < 0.001$ for configurations 2,4,5, $p = 0.028$ for configuration 1, where the failed trials were not discarded).

Removing plasticity in the connections from $A$ to $C$ or from $A$ to $M$ individually had for the most part no statistically significant effects. Removing plasticity in both connections simultaneously, however, roughly duplicated the error in configurations 2 and 4 during center-out reaching (one-sided t-test, $p < 0.001$). Error may be slightly increased for configuration 1 (the small sample size allowed no strong conclusions), whereas configuration 5 was seemingly unaffected.

Configurations (1,2,4,5) could still learn to reach after removing the $ACT$ unit. Configuration 4 roughly duplicated its center-out reaching error ($p < 0.001$) and its time to initially reach ($p = .016$). Configuration 1 increased its time to initially reach about 3 times ($p < 0.001$), and the other two configurations were seemingly unaffected. The $ACT$ unit was essential for previous, less robust versions of the model.

Removing noise made learning too slow, to the point where mean error in the last 4 training reaches could not be reduced below 10 cm in any trial for configurations 4, and 5. It was reduced below 10 cm in a single trial for configuration 2. Configuration 1 managed t learn normally. Center-out reaches were not possible in configurations 2, 4, and 5 with mean errors of ($15 \pm 8 | 17 \pm 11 | 9 \pm 3$) centimeters respectively, at least 3 times larger than the models with noise ($p < 0.001$). Center-out reaches were normal in configuration 1, but the first reach with mean error below 10 centimeters took significantly longer to happen (from 2.5 to 6.4 attempts in average, $p = 0.001$).

Removing Ia and Ib afferents, and instead sending the output of II afferents to $C$, $M$, and $S_A$ prevented reaching in configurations 1, 2, 3, and 4 (except for a single trial in configuration 1). Configuration 5 could still learn to reach, but the mean error in the last 4 training reaches and during the center-out reaching was significantly higher ($p < 0.001$).

Removing the agonist-antagonist connections in $C$ prevented reaching in (5| 0| 0| 3| 2) trials for configurations (1,2,3,4,5) respectively. Error in center-out reaching was significantly increased for configurations 4 and 5, and it did not increase significantly for configurations 2 and 3.

*Appendix 1—figure 2* is the same as *Figure 2* of the main text, but in this case the noise and the ACT units were both removed. The average distance from the hand to the target was roughly 18 cm. A video illustrating this failure to learn is included with the supplementary videos.

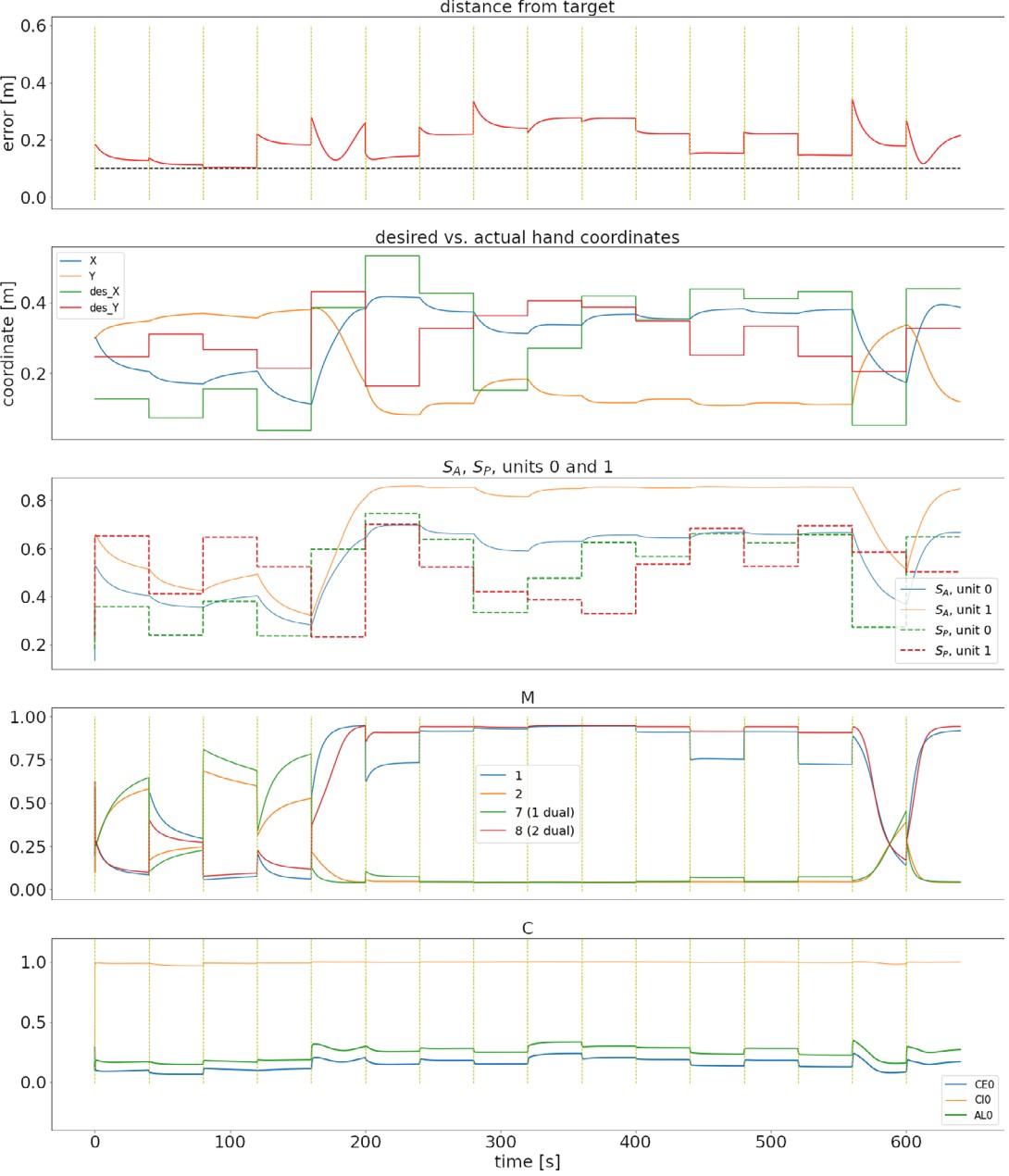

**Appendix 1—figure 2.** A failure to learn for a simulation of configuration 1 (spinal learning) with no noise and no ACT unit. Captions are as in *Figure 2* of the main text.

Configuration 1 was resilient to removal of noise and the ACT unit individually. The simulation of *Appendix 1—figure 2* suggests that removing more than one element will have larger consequences on the performance of the model.

## Comparison of the 5 configurations

Once again, the 5 configurations in this paper are represented by a number: 1=spinal learning, 2=cortical learning, 3=static network, 4=synergistic network, 5=mixed errors network. The following table shows the connectivity in each one. Abbreviations: A2A: all-to-all, O2O: one-to-one, DH: the differential Hebbian learning rule from *Verduzco-Flores et al., 2022*, IC: the Input Correlation learning rule, S: static connections (see section for details on the weights of static connections).

| Configuration | $S_{PA}$ to $M$ | $M$ to $C$ | $A$ to $C$, $M$ |
|---|---|---|---|
| 1 | O2O, S | A2A, DH | A2A, IC |
| 2 | A2A, DH | S | A2A, IC |
| 3 | O2O, S | S | S |
| 4 | O2O, S | A2A, DH | A2A, IC |
| 5 | S | A2A, DH | A2A, IC |

- The spinal learning model (configuration 1) is a "basic" network where the input-output structure of the control loop happens in the spinal cord, in the connections from M to C.
- The cortical learning model (configuration 2) is also a "basic" network, but the input-output structure is resolved in the intracortical connections from $S_{PA}$ to $M$.
- The static network (configuration 3) uses only static connections, and is meant to show that the results in the paper appear in a close to optimal configuration of feedback control, rather than being some sophisticated product of the plasticity rules.
- The synergistic network (configuration 4) is an extension of configuration 1, where the spinal cord has 12 CE, CI pairs rather than 6, and each pair stimulates 2α motoneurons.
- The mixed errors network (configuration 5) is a different variation of configuration 1, where the connections from $S_{PA}$ to $M$ are not one-to-one, but instead come from an orthogonal matrix.

The following table summarizes the numerical results for the 5 configurations.

| Measurement | 1 | 2 | 3 | 4 | 5 |
|---|---|---|---|---|---|
| Failed reaches[1] | $1.8 \pm 2$ | $1.2 \pm .9$ | $0 \pm 0$ | $1.6 \pm 1.3$ | $4 \pm 2.5$ |
| Center-out error[2] | $3.3 \pm .01$ | $2.9 \pm .001$ | $2.9 \pm .0003$ | $3 \pm .008$ | $2.8 \pm .0007$ |
| $M$ units tuned | | | | | |
| to direction | $11.8 \pm .4$ | $12 \pm 0$ | $12 \pm 0$ | $12 \pm 0$ | $12 \pm 0$ |
| $R^2$ forfor | | | | | |
| predicted PD | $.74 \pm .18$ | $.88 \pm .14$ | $.86 \pm .01$ | $.89 \pm .06$ | $.82 \pm .03$ |
| PD distribution | | | | | |
| main axis (deg) | $59 \pm 7$ | $52 \pm 2$ | $54 \pm .5$ | $60 \pm 3$ | $58 \pm 1$ |
| PD drift | | | | | |
| angle (deg) | $3.3 \pm 2.4$ | $4.9 \pm 2.1$ | $.3 \pm .2$ | $1.8 \pm 1.3$ | $7 \pm 6$ |
| Muscle PD | | | | | |
| drift angle (deg) | $3.8 \pm 2.1$ | $6.4 \pm 2.9$ | $.2 \pm .2$ | $11.4 \pm 15.2$ | $27.7 \pm 34.5$ |
| Center-out error | | | | | |
| (10 targets) | 3 | 3.6 | 2.9 | 2.6 | 4.5 |
| Variance in | | | | | |
| first jPCA | $.42 \pm .04$ | $.42 \pm .04$ | $.46 \pm .03$ | $.45 \pm .04$ | $.47 \pm .07$ |
| Center-out error | | | | | |
| (light arm) | 2.5 | 3.2 | 3 | 6.1 | 2.9 |
| Center-out error | | | | | |
| (heavy arm) | 2.4 | 3.3 | 2.9 | 5.6 | 3.2 |

Average number of reaches before the first reach when the mean error was below 10 cm. [2] Average distance (in centimeters) between the hand and the target during center-out reaching.

## Gain and oscillations

The gain of a feedback loop describes how the amplitude of the error signal increases as it gets transformed into a control signal sent to the plant. As described in the Methods (section), we used a relatively low number of iterations of an optimization algorithm to find suitable parameters for each configuration of the model. This led to configurations with gains that were coarsely tuned. *Figure 3—figure supplement 1* is analogous to *Figure 3*, and shows the hand trajectories right after the optimization algorithm was finished. It can be observed that configurations 2 and 3 were particularly prone to oscillations, and configuration 1 would undershoot many targets.

To improve performance, as well as to facilitate comparison of the 5 configurations, we adjusted their gains. This involved manually adjusting the slope of the sigmoidal units in populations $M$ and $S_A$, until they appeared stable, but on the verge of oscillating (so reaching would be faster). This required from 1 to 3 attempts. The gain of configuration 1 was slightly increased, whereas the gain of configurations 2,3,4 was reduced. Configuration 5 was left with the same parameters.

The trajectories in panels C and D of *Figure 3—figure supplement 1* are reminiscent of terminal tremors in cerebellar ataxia. An animation showing the movement of the arm for the 5 configurations before gain adjustment is included among the Supplementary Videos. In addition, *Figure 3—figure supplements 2 and 3* show the error and activity of several units during the training reaches for configurations 3 and 4, analogous to *Figure 2*. It can be observed that the oscillations are present in the whole network, suggesting that the control signals are trying to catch up with an error that keeps reversing direction.

## Supplementary videos

To help visualization of the arm's learning and performance under different conditions, 4 videos were produced. The videos indicate the model configuration using the enumeration from this Appendix: 1=spinal learning, 2=cortical learning, 3=static network, 4=synergistic network, 5=mixed errors network.

To download these videos, please visit https://gitlab.com/sergio.verduzco/public_materials/-/tree/master/adaptive_plasticity

The videos' content is as follows:

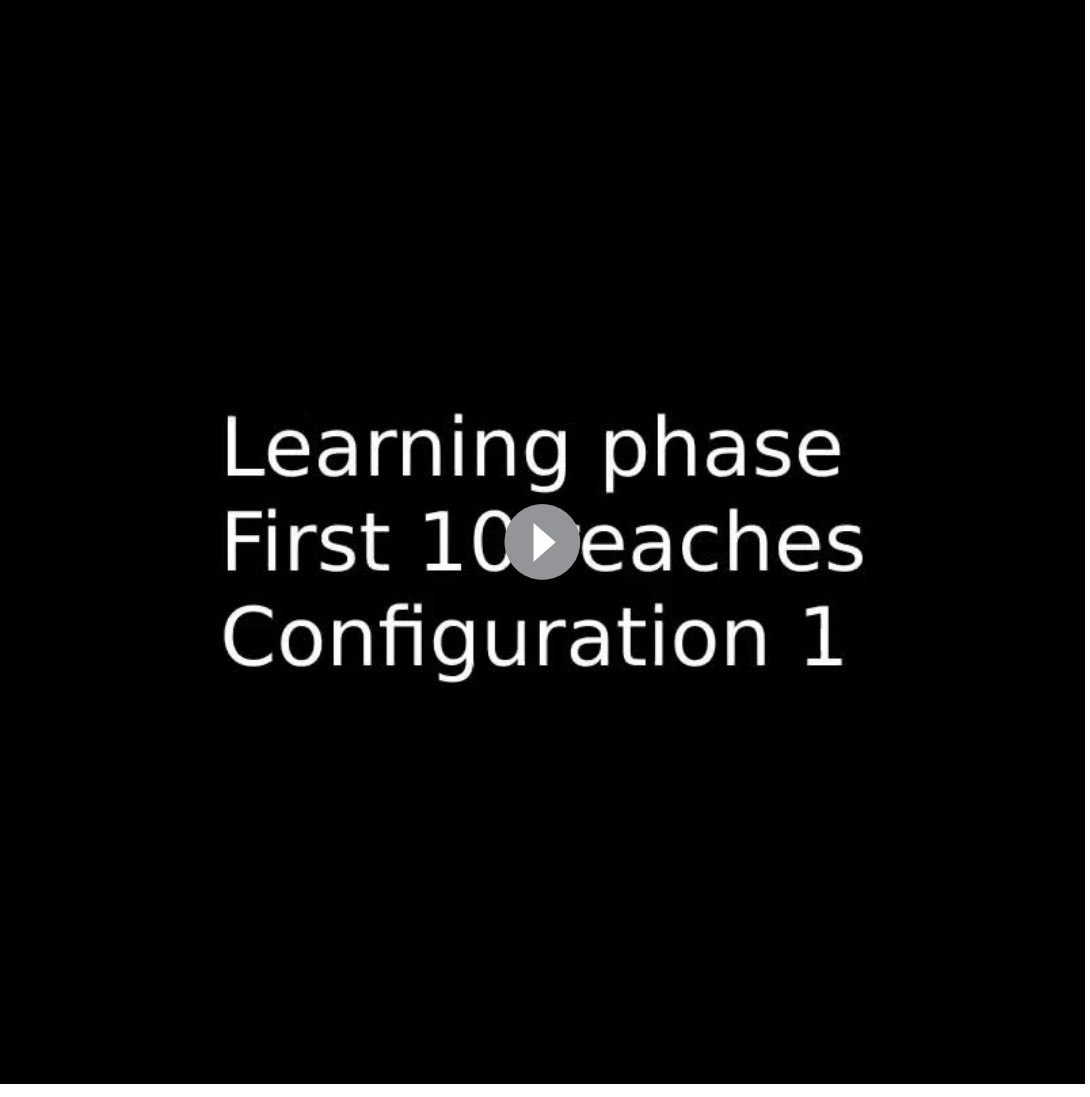

**Appendix 1—video 1.** Visualization of the arm and the muscles during the learning phase for configuration 1. Data comes from the simulation shown in Figure 2. Speed is roughly 4 X.

https://elifesciences.org/articles/77216/figures#video1

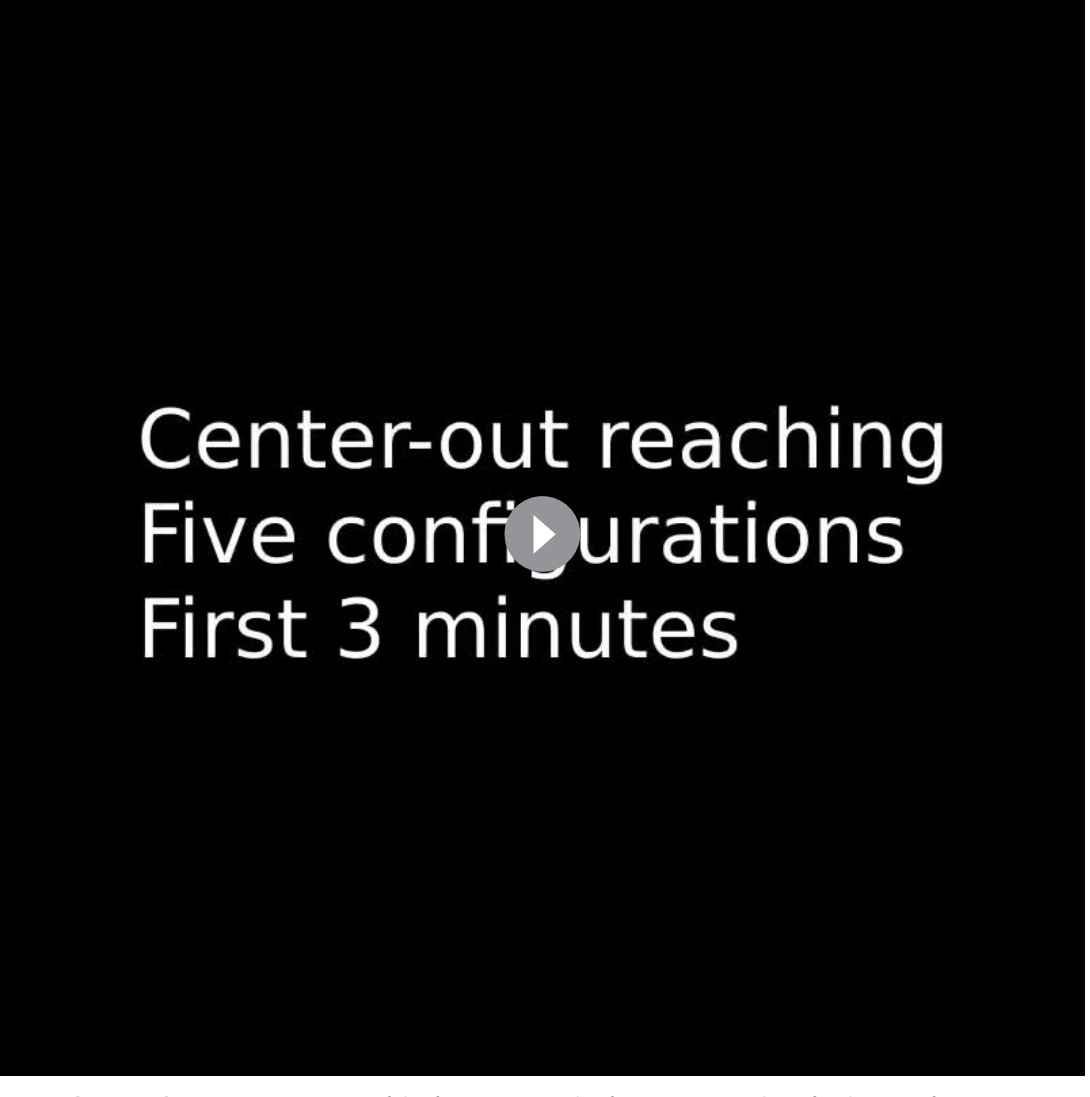

**Appendix 1—video 2.** Arm animation of the first 180 seconds of center-out reaching for the 5 configurations. Speed is roughly 4 X.

https://elifesciences.org/articles/77216/figures#video2

**Appendix 1—video 3.** The learning phase for a simulation with configuration 1. Both noise and the unit were removed, reducing exploration and disrupting learning. Data comes from the same simulation in Appendix 1—figure 2. Speed is 4 X.

https://elifesciences.org/articles/77216/figures#video3

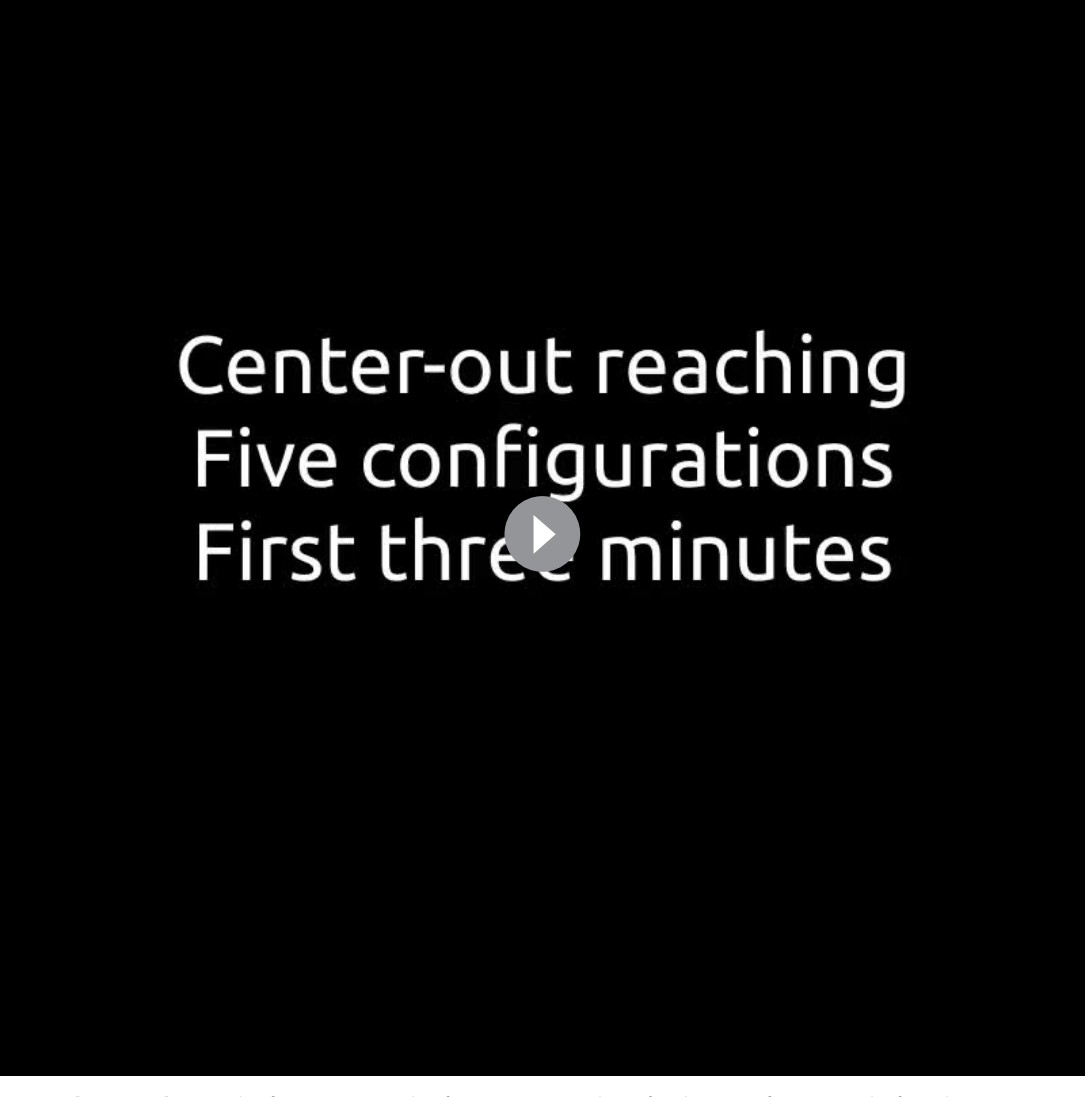

**Appendix 1—video 4.** The first 180 seconds of center-out reaching for the 5 configurations before the gains were adjusted. Speed is roughly 4 X. Configuration 2 shows target-dependent oscillations after 70 seconds.

https://elifesciences.org/articles/77216/figures#video4

## Parameter values

Values appear in order for configurations 1–5. A single number means that all configurations use that same value.

### Unit parameters

The superscript $x$ on a population name indicates that a parameter has heterogeneous values. This means that a random value was added to the parameter for each individual unit. This random value comes from a uniform distribution centered at zero, with a width equal to 1% of the reported parameter value.

| Parameter | Equation | Population | Value |
| --- | --- | --- | --- |
| $\tau_u$ | 5 | $CHG, A, ACT$ | 10 [ms] |
| | | $\alpha, S_A, S_{PA}^x, CI$ | 20 [ms] |
| | | $CE$ | 140, 70, 150, 180, 110 [ms] |

*Continued on next page*

*Continued*

| Parameter | Equation | Population | Value |
|---|---|---|---|
|  |  | $M^x$ | 50 [ms] |
| $\beta$ | 6 | $\alpha^x, ACT$ | 2 |
|  |  | $CE$ | 1.63, 1.72, 1.70, 3.38, 1.44 |
|  |  | $CI$ | 4.0, 3.38, 3.44, 2.46, 3.63 |
|  |  | $M^x$ | 1.5, 1.5, 2.0, 2.0, 1.17 |
|  |  | $S_A$ | 3.0, 2.2, 2.0, 2.3, 3.0 |
|  |  | $CHG, S_{PA}^x$ | 9 |
| $\eta$ | 6 | $\alpha^x$ | 1.1 |
|  |  | $CE$ | 2.0, 1.93, 2.13, 2.31, 1.67 |
|  |  | $CI$ | 1.5, 1.41, 1.63, 1.72, 1.7 |
|  |  | $M_x$ | 1.3, 1.96, 0.68, 1.19, 1.38 |
|  |  | $ACT$ | 1 |
|  |  | $S_A$ units 0,3 | 0.75 |
|  |  | $S_A$ units 1,2,5 | .4 |
|  |  | $S_A$ unit 4 | .3 |
|  |  | $CHG$ | .25 |
|  |  | $S_{PA}^x$ | .1 |
| $\varsigma$ | 7 | $CE, CI$ | 0.63, 0, 0, 0.69, 0.72 |
|  |  | $M$ | 0, 0.62, 0, 0, 0 |
| $\tau_a$ | 8 | $A$ | 10 [ms] |
| $T$ | 8 | $A$ | .2 |
|  |  | $I_b, II$ afferents |  |
|  |  | $A$ | 0 |
|  |  | $I_a$ afferents |  |
| $\tau_{slow}$ | 12, 13 | $CE, CI$ | 11 [s] |
| $\theta_{ACT}$ | 15, 16 | $ACT$ | .31 |
| $\tau_{ACT}$ | 16 | $ACT$ | 10 [ms] |
| $\gamma$ | 16 | $ACT$ | 8 |
| $\alpha$ | 17 | $CHG$ (synapse) | 20 |

## Learning rules

| Parameter | Equation | Value |
|---|---|---|
| $\Delta t$ | 9 | 0.33, 0.37, 0.15, 0.36, 0.32 [s] |
| $\alpha(M \text{ to } C)$ | 10 | 500, 0, 0, 500, 500 |

*Continued on next page*

*Continued*

| Parameter | Equation | Value |
|---|---|---|
| $\alpha(S_{PA} \text{ to } M)$ | 10 | 0, 527, 0, 0, 0 |
| $\alpha(M \text{ to } \alpha)$ | 10 | 300, 0, 0, 300, 300 |
| $\lambda$ | 10 | .03 |
| $\omega_{sa}(M \text{ to } C, \alpha)^1$ | 10 | 2.52 |
| $\omega_{sa}(S_{PA} \text{ to } M)$ | 10 | 3.23, 3.19, 2.98, 3.23, 3.23 |
| $\omega_{sb}(M \text{ to } C)$ | 10 | 3.29, 2.14, 1.50, 3.69, 0.57 |
| $\omega_{sb}(M \text{ to } \alpha)$ | 10 | 2.86, 2.86, 1.50, 2.86, 2.86 |
| $\alpha_{IC}(A \text{ to } M)$ | 11 | 26.17 |
| $\alpha_{IC}(A \text{ to } C)$ | 11 | 22.5 |
| $\omega_s(A \text{ to } M)$ | 11 | 0.85, 1.14, 1, 0.53, 0.53 |
| $\omega_s(A \text{ to } C)$ | 11 | 1.68, 2, 2, 1.55, 2.88 |
| $\omega_{max}(A \text{ to } M)$ | 11 | .48,.22,.2,.25,.33 |
| $\omega_{max}(A \text{ to } C)$ | 11 | .3,.64,.64,.28,.59 |

1 Constraints in the sum of weights are also used with static connections

## Muscles and afferents

| Parameter | Equation | Value |
|---|---|---|
| $K_{SE}$ | 18 | 20 [N/m] |
| $K_{PE}$ | 18 | 20 [N/m] |
| $b$ | 18 | 1 [N.s / m] |
| $g$ | 18 muscles 0,3 | 67.11 [N] |
| $g$ | 18 muscles 1,2,4,5 | .75 [N] |
| $K_{SE}^s$ | 19,20,21 | 2 [N/m] |
| $K_{PE}^s$ | 19,20,21 | 2 [N/m] |
| $b^s$ | 19, 21 | .5[Ns / m] |
| $K_{SE}^d$ | 19,20,21 | 1 [N/m] |
| $K_{PE}^d$ | 19 | .2 [N/m] |
| $b^d$ | 19 | 2[N. s / m] |
| $l_0^s$ | 19 | .7 |
| $l_0^d$ | 19 | .8 |
| $g_{I_a}$ | 20 muscles 0–5 | $[7.5, 25, 25, 7.5, 25, 25][m^{-1}]$ |

*Continued on next page*

*Continued*

| Parameter | Equation | Value |
|---|---|---|
| $f_s^{I_a}$ | 20 | 0.1 |
| $g_{II}$ | 21 | [5.46, 8, 8, 5.46, 8, 8]*, |
| | muscles 0–5 | [5.8, 8, 8, 5.8, 8, 8]** $[m^{-1}]$ |
| $f_s^{II}$ | 21 | 0.5 |
| $g_{I_b}$ | 22 | 1 |
| $T_0$ | 22 | 10 [N] |
| $\tau_g$ | 23 | 50 [ms] |

## Connection delays and weights

Connections considered "local" used a delay of 10 [ms], unless those those connections implied a not-modeled disynaptic inhibition. All other connections had a delay of 20 [ms].

| Source | Target | Delay |
|---|---|---|
| $A$ | $M, S_A$ | 20 [ms] |
| $ACT$ | $CE, CI$ | |
| $\alpha$ | muscle | |
| $M$ | $CE, CI, \alpha$ | |
| $M$ | $M$ | |
| afferents | $A$ | |
| $CHG, S_{PA}$ | $ACT$ | |
| $S_{PA}$ | $M, S_{PA}$ | |
| $A$ | $M$ | 10 [ms] |
| $CE, CI$ | $\alpha$ | |
| $CE, CI$ | $CE, CI$ | |
| $CI$ | $CE$ | |
| $S_A, S_P$ | $S_{PA}$ | |
| $S_P$ | $CHG$ | |

The next table shows the value of fixed synaptic weights not specified in section. Columns indicate the source of the connection, rows indicate the target. "Aff" stands for the muscle afferents. Potentially plastic connections are marked as "+". If a connection marked "+" is static in one of the configurations, its weight is determined by the $\omega_{sa}, \omega_{sb}, \omega_s$ parameters of *Equations 10, 11*.

| | $\alpha$ | $A$ | $CE$ | $CI$ | $M$ | **Aff.** | $S_A$ | $S_P$ | $S_{PA}$ |
|---|---|---|---|---|---|---|---|---|---|
| $\alpha$ | | 1 | -1 | | + | | | | |
| $A$ | | | | | | 2 $(I_a, I_b)$, 4 $(II)$ | | | |
| $ACT$ | | | | | | | | | 1 |
| $CE$ | | + | $.5^a, .18^b$ | $-1.8^c$ | + | | | | |
| $CHG$ | | | | | | | | + | |

*Continued on next page*

*Continued*

|  | $\alpha$ | A | CE | CI | M | Aff. | $S_A$ | $S_P$ | $S_{PA}$ |
|---|---|---|---|---|---|---|---|---|---|
| *CI* |  | + | $.5^c$ |  | + |  |  |  |  |
|  |  |  | $1.83^d, .16^e$ |  |  |  |  |  |  |
| *M* |  | + |  |  | * |  |  |  | + |
| muscles | 1 |  |  |  |  |  |  |  |  |
| $S_A$ |  | 1 |  |  |  |  |  |  |  |
| $S_{PA}$ |  |  |  |  |  |  | 1 or –1 | 1 or –1 | –1.77 |

a Agonist connections, b Partial agonist connections. c Withing the same triplet. d Antagonist connections. e Partial antagonist connections.

units inhibited their duals with weights that depended on the configuration: −0.93,−0.74, −1.00,−1.14, 0.0.

All connections whose source is *CHG* or *ACT* have a weight of 1.

