## [Editor Report]

This solid modelling study presents a valuable contribution toward understanding the neural control of movement. The authors show that a minimal model comprising key sensorimotor cortical areas as well as a spinal circuits controlling a limb readily replicates landmark observations from behavioural and electrophysiological studies. This work will be of broad interest to motor control researchers, as well as to neurophysiologists interested in testing the predictions derived from this model.

---

## [Decision Letter]

**Decision letter after peer review:**

Thank you for submitting your article "Adaptive plasticity in the spinal cord can produce reaching from scratch and reproduces motor cortex directional tuning" for consideration by *eLife*. Your article has been reviewed by 3 peer reviewers, and the evaluation has been overseen by a Reviewing Editor and Tamar Makin as the Senior Editor. The following individuals involved in the review of your submission have agreed to reveal their identity: Marco Capogrosso (Reviewer #2); Rune W Berg (Reviewer #3).

The reviewers have discussed their reviews with one another, and the Reviewing Editor has drafted this to help you prepare a revised submission. The general consensus was that this is an interesting model, which will be a valuable contribution to the literature provided that additional controls and clarifications are provided.

Below you will find the individual reviewers comments. In addition to these, the editor prepared for the authors a summary of the essential revisions required. This is based on both the individual comments and the extensive following discussion between the reviewers and editor.

Essential revisions:

1. The paper focuses on an intriguing new hypothesis: that the spinal cord minimizes an error signal generated in motor cortex. Given the novelty of this hypothesis and its implications for the present manuscript, it is critical that the authors provide additional controls/analyses that specifically address it. These include:

a) What happens if error minimisation happened in the motor cortex rather than in the spinal cord; would the model still produce reaching movements? Do the cortical neurons look like actual neurons (distribution of directional tuning, rotational dynamics)? Do "synergies" still emerge in the spinal cord?

b) Does a model with the same hierarchical structure that doesn't perform error minimisation and is still capable of reaching exhibit the properties that the current model does (in terms of motor cortical activity, synergies, etc)?

2. The authors posit that motor cortex reports and error that is being minimised during learning. Under this assumption, the motor cortical signals should be reduced to zero, which is what indeed seems to happen, e.g., in Figure 2E. Nevertheless, it is known that the activity of motor cortical neurons is not zero during the execution of a correct movement, even in highly trained individuals: e.g., trained monkeys have rotation dynamics when reaching (Churchland Cunningham et al. Nature 2012) or hand cycling (Russo et al. Neuron 2018). How can these observations be reconciled with the current model?

3. The cerebellum is mentioned a couple of times but not included in the model. Based on the known fact that the cerebellum receives and integrates both massive corticopontine input and spinocerebellar proprioceptive input, its role is hence often assumed to be exactly what the motor cortex does in this model. The rationale behind not letting Cerebellum assume this role, but instead making the motor cortex assume this role, is not clear.

4. The authors claim that previously known phenomena (convergent force fields and rotational dynamics) "emerge" with their learning model. However, this is not supported by the results. Firstly, when they tested the convergent force field, the spinal cord was separated from the descending and ascending connections with cortex. This means that this observed property is independent of having learned the cortico-spinal connections. In addition, rotational dynamics were almost exclusively observed in the one-to-one cortical coupling, which the authors call a "less-plausible model. Therefore, these phenomena are not an emergent property that results from their learning model. It is necessary to specify what are the essential factors for each observed phenomenon.

5. This model has attributed all of the learning effects to the synaptic connection between the cortex and spinal cord (descending and ascending connections). However, it has been repeatedly demonstrated that plastic changes within the cortex occur during motor learning (Roth et al. Neuron 2020). How is the physiological relevance of attributing all the learning to descending and ascending pathways justified?

6. In relation to the previous comment: an equivalent circuit could be achieved by changing the intracortical synaptic connections. Why is the corticospinal connection the sole target of learning, and how do the results change (or remain unchanged) if learning happens in other parts of the circuit?

7. No model can reproduce the whole CNS. However, the reviewers think that a few key features should be added to the model:

a) The authors have not included any form of feedback in the spinal cord. Indeed, short-latency stretch reflexes are critical for movement: coupled with motoneurons, they are the basic units to generate movement (Xu, Tianqui et al. PNAS 2018); models of human walking show that gait can emerge only considering sensory inputs to the cord (Song and Geyer J Physiol 2015); and their absence prevents recovery from spinal cord injury (Takeoka, Aya et al. Cell 2014). Ia afferents have perhaps the strongest monosynaptic glutamatergic synaptic input to spinal motoneurons and constitute a primary driver of spinal motoneuron excitability which this model is currently missing. Does adding short-latency feedback change the conclusion of the paper?

b) As the authors discussed in the manuscript, the model omits many components that are known to be involved in learning and, as a result, the model reproduces the impaired motor control rather than the intact state. However, this prevents us from verifying whether the model properly reproduced the biological motor control. Specifically, they claim that the model reproduced ataxic reaching which resemble to the cerebellum patients. However, it is not surprising that a poorly tuned feedback controller (i.e. PID controller) shows an oscillation. Therefore, it is necessary for the authors to show that their model significantly resembles the biological motor control by comparing to the model which share the hierarchical structures but the error is corrected within the cortex and only motor commands descend to the spinal cord.

c) 20 % of corticospinal projections to motoneurons innervating the arm and particularly the hand are thought to be monosynaptic. Does their addition change the present results?

d) The proportions and the connectivity of the network make the model seem less biologically plausible, which should at least be addressed in the text. First, the number of neurons in the populations at various levels seems rather small compared to the immense size of these structures. Why not have more neurons in the networks? This applies to all parts of the model, also to the spinal network. In many species, the ratio of motor neurons to interneurons is about 1:10, yet in this model, it is more like 1:2 (the trio). Also, the connectivity has a bias towards the feedforward network, yet local recurrent connectivity seems to be widely present in the nervous system. The proportions in the connectivity from M to C could also be problematic. All the spinal units (c = [c1, …. ,cN]) receive commands/error signal from M (e = [e1, …. ,eM]). Is M>N? Presuming M is > N this would represent a converging input from M to C, and that is problematic since the corticospinal projections are rather sparse. More likely there is diverging connectivity from M to C. It is unclear to me if this is an important problem, it may not be, but it would be appropriate to address such proportions in the text.

e) The model does not include any form of "pattern generator", a limitation the authors are aware of, and denote as the "supraspinal pattern formation" problem. They seem to export this to another area of the brain while the motor cortex-spinal cord loop takes care of an appropriate execution. This needs to be clarified and/or addressed with the model.

f) It is well-known that the spinal cord autonomously can produce motor behaviors. It is unclear whether C in the model is capable of this.

g) Regarding the synergistic model, their model has spinal interneurons recruiting two motoneuron pools, which is very different from the muscle synergies observed in animals (d'Avella et al., J Neurosci 2006). It is necessary to justify why this type of synergies should be used here.

*Reviewer #1 (Recommendations for the authors):*

The authors aimed to study corticospinal control of reach movements. More specifically they aim to understand how an unsupervised system could learn to reach and whether that system would then show properties observed in experiments. For this, the authors conceived a neural network model with defined hierarchy and architecture coupled to a simple biomechanical arm model in a closed-loop that is capable of learning a reach-out task. Interestingly, the model shows several properties found in experimental data such as directional tuning of motor cortical units as well as discrete force fields in the spinal cord.

Strengths

The paper has many strengths. It is rigorous and well written, and while not well articulated in the introduction I completely agree with the authors that computational models are necessary if we want to aim at any variation of the concept of "understanding" in relation to the central nervous system. Therefore their goal is absolutely significant and relevant.

I also particularly liked the approach to start from specific hypotheses on the structure and function of the different modeled layers while leaving completely free the organization and strength of synapses in the system. This approach is elegant because starting from clear hypotheses it allows to observe what properties emerge "spontaneously" and compare these properties with experimental data.

The results obtained are reasonable in terms of kinematics and it's interesting to observe the emergence of these properties from an error minimization rule imposed within the spinal cord.

Weaknesses

The conclusions that the authors make in the paper are very strong. Given the emergence of certain interesting properties within the nervous system, it is implicit the implication that this model can have some sort of general meaning, and most importantly, that it validates from the theoretical point of view the homeostatic control idea. More specifically, the paper starts with a very strong hypothesis, which is that the spinal cord is minimizing an error signal generated in motor cortex. For an experimental neuroscientist (as myself) this is intriguing but also strikes as a very strong interpretation of the role of what we call motor cortex that for decades has been imagined as an actuator of movements. Because of the importance of the implication I don't think that the properties shown in the paper are sufficient to make such a claim, even if implicit. Therefore I think that some form of control is required. I understand completely that models are hard to build, but I can't escape wondering what would happen if the motor cortex would not be producing an error signal. What if that error minimization happened instead in the motor cortex and not in the spinal cord? Would your model still produce reach movements? Would those cortical neurons show directionality? Would "synergies" still emerge in the cord? I think this is an important point if we're claiming that those properties emerge as a consequence of the fact that the motor cortex is calculating an error and the spinal cord is minimizing that error. Therefore, to validate the main hypothesis of the paper, which is that the motor cortex is calculating an error and the spinal cord is minimizing that error, then I would like to see that a model that does not do that, but it's still capable of reaching while maintaining the same hierarchical architecture would not show the properties that the authors found in their model.

The second weakness is somehow related to the first. There are certain assumptions on the structure that are not true experimentally and it would be important to incorporate these because they may change the observed properties and could thus be used as further validation.

The first missing property is direct cortico-spinal control of spinal motoneurons. The authors say in the introduction that the motor cortex does not activate spinal motoneurons directly and only connects to interneurons. Instead, the major difference between primates (including humans) and all the other animals is precisely the existence of direct monosynaptic connections between the motor cortex and muscles of the upper limb (particularly the hand). These monosynaptic components are thought to represent about 20% of the cortico-spinal tract. It would be important to see whether their presence changes the results that they obtained.

The second more important functional inaccuracy concerns the spinal cord. The authors send their sensory signals to the "thalamus" which then relays them to the "sensory cortex" and "motor cortex". In the supplementary material they described the fact that they decided not to include what they call "short reflex" that would connect "the thalamus" to the spinal cord, or provide a feedback in some way. Therefore in their model, the spinal cord is not receiving a feedback at all. Unfortunately, this is in stark contrast with the actual structure of the spinal cord. Proprioceptive afferents are pivotal members of the spinal motor infrastructure. They are so important than in simple animals, coupled with motoneurons, they constitute the units that generate movement (Xu, Tianqi, et al. "Descending pathway facilitates undulatory wave propagation in *Caenorhabditis elegans* through gap junctions." Proceedings of the National Academy of Sciences 115.19 (2018): E4493-E4502.); models of human locomotion show that it is possible to build walking models without pattern generators that only consider sensory inputs int he spinal cord (Song, S., and Geyer, H. (2015). A neural circuitry that emphasizes spinal feedback generates diverse behaviours of human locomotion. J. Physiol. 593, 3493-3511.) and they seem to be critical to direct motor learning and recovery after spinal cord injury (Takeoka, Aya, et al. "Muscle spindle feedback directs locomotor recovery and circuit reorganization after spinal cord injury." Cell 159.7 (2014): 1626-1639.) Most importantly Ia afferents have perhaps the strongest monosynaptic glutamatergic synaptic input to the spinal motoneurons and constitute a primary driver of spinal motoneuron excitability which this model is currently missing. Additionally, some of the agonist-antagonist relationship that the authors find could be significantly changed by the presence of these afferents because Ia afferents connect not only to homologous motoneurons but up to 60% of motoneurons of synergistic muscles and they inhibit antagonists via Ia inhibitory interneurons (See Moraud E. (2016) Mechanisms Underlying the Neuromodulation of Spinal Circuits for Correcting Gait and Balance Deficits after Spinal Cord Injury, Neuron).

In consequence, the presence of this "short-feedback" seems paramount to me to have an accurate representation of the spinal cord, and it's important to verify that this structure would not change the main conclusions of the paper.

Indeed, as the authors have correctly implied, the main difference between the nervous system and an artificial neural network is the underlying defined architecture and hierarchy that must be represented at least for the most important elements. And spinal sensory feedback is a building block of movement and learning. Indeed, completely paralyzed rats are capable of learning different walking patterns when totally disconnected from the brain if excitation is provided by means of electrical stimulation and likely sensory afferents are the driver of this learning (Courtine, Grégoire, et al., "Transformation of nonfunctional spinal circuits into functional states after the loss of brain input." Nature neuroscience 12.10 (2009): 1333-1342.).

*Reviewer #2 (Recommendations for the authors):*

In this study, the authors investigate how the motor cortex and spinal cord interact in a long-loop reflex organization, from a computational modeling perspective. The motor cortex represents kinematic aspects, i.e. the actual movement of the hand and arm, as opposed to dynamics, which would involve the forces exerted by the muscles or higher-level parameters of the movement. Nevertheless, to achieve the end goal of a certain movement of the hand, a transformation from actuator (muscle) coordinates has to take place, and this is a challenging process. The authors propose that the motor cortex report an error between the desired and the actual movement, use computational modeling to explain how a flexible error reduction mechanism can be achieved and explained by computational modeling of the interplay between cortex and the spinal cord.

Strengths: A theoretical analysis of the interplay between the spinal cord and motor cortex has rarely been studied previously especially using the long-loop reflex, dynamics, pattern formation, and plasticity. This theoretical contribution is important, especially in the light of the vast new experimental data that is being generated by e.g. calcium imaging and electrophysiology and the new observations of rotational cortical dynamics. It is also rare that the role of the spinal cord is being incorporated into the executing of movement in parallel with the motor cortex.

Weaknesses: Some of the assumptions hinges on a flexible error reduction mechanism, which is described in a previous (unpublished) study, which seems somewhat fragile. The model does not really get to the root of the problem of how the nervous system generates the desired movement, which the authors are aware of and call the "supra spinal pattern formation" problem. They seem to export this problem to another area of the brain and describe how the motor cortex-spinal cord loop takes care of an appropriate execution.

The study evaluates multiple aspects of the model in comparison with experimental observation such as direction tuning curve, rotational dynamics, and learning.

Overall, this is an interesting and well-executed study.

Specific comments:

There are a couple of issues regarding the model that is not so clear (or perhaps well-hidden somewhere in the supplement). It is stated that the role of the motor cortex is to report the "error signal" of the movement, i.e. the signal coming from SPA, which is the difference between the desired movement (SP) and the actual movement (SA). First, if the motor cortex is reporting the error, and this error is being minimized while learning, then the signal in the motor cortex should ideally be reduced to zero, which is also what seems to happen in e.g. Figure 2E. Nevertheless, it is known that activity in the motor cortex is not zero during the execution of correct movement, especially in highly trained individuals, e.g. trained monkeys have rotational dynamics when hand-cycling (Churchland 2012). This also seems to be the case in the current study with directional tuning (Figure 5) and rotation, which appear contradictory to me. This is a bit unclear how this is possible. Second, the cerebellum is mentioned a couple of times but not included in the model. It is known that the cerebellum receives and integrates both massive corticopontine input and spinocerebellar proprioceptive input and the role of the cerebellum is hence often assumed to do exactly what the motor cortex does in this model. The rationale behind not letting Cerebellum assume this role, but instead making the motor cortex assume this role, is not clear to me.

The current model does not explain what the authors call the "supra spinal pattern formation" problem, i.e. how the motor commands arise in the neuronal networks in the cortex and elsewhere. The problem of motor commands is conveniently exported to the external parameter "SP" module that is generating the intended movement. This is for a good reason since it is indeed a hard problem to explain how motor programs arise. I do not see a solution to this issue, only that it is a limitation to be aware of, which the authors seem to be. Nevertheless, the model proposed by the authors makes an effort to explain some of the dynamics of the different regions involved in properly learning and executing the motor commands.

I have some concerns regarding the network, the proportions, and the connectivity that make the model seem less biologically plausible, which should at least be addressed in the text. First, the number of neurons in the populations at various levels seems rather small compared to the immense size of these structures. Why not have more neurons in the networks? This applies to all parts of the model, also to the spinal network. In many species, the ratio of motor neurons to interneurons is about 1:10, yet in this model, it is more like 1:2 (the trio). Also, the connectivity has a bias towards the feedforward network, yet local recurrent connectivity seems to be widely present in the nervous system. The proportions in the connectivity from M to C could also be problematic. All the spinal units (c = [c1, …. ,cN]) receive commands/error signal from M (e = [e1, …. ,eM]). Is M>N? Presuming M is > N this would represent a converging input from M to C, and that is problematic since the corticospinal projections are rather sparse. More likely there is diverging connectivity from M to C. It is unclear to me if this is an important problem, it may not be, but it would be appropriate to address such proportions in the text.*Reviewer #3 (Recommendations for the authors):*

This study proposes a new computational model for learning upper limb reaching control. In particular, this study develops a model that satisfies biological plausibility, which has not always been satisfied by previous models: (1) a network of the whole sensorimotor loop, (2) a model using neural elements, (3) learning using local information, and (4) continuous online learning. Specifically, their model showed that learning in the synaptic weights of descending and ascending projections between the cortex and spinal cord can reproduce ataxic reaching often observed in cerebellar patients. Furthermore, various phenomena that have been demonstrated in the motor control field emerge with this model. Based on these results, the authors argue that this is a biologically plausible learning model for animals' reaching control.

Strength:

1) This study examines a number of previous motor learning models and defines the constraints to develop a biologically plausible model. Given that there are too many parameters to be considered to build the motor control model, I truly appreciate the authors defining the constraints for a biological motor learning model and clarifying the research goal.

Weakness:

1) As the authors discussed in the manuscript, the model omits many components that are known to be involved in learning and, as a result, the model reproduces the impaired motor control rather than the intact state. However, this prevents us from verifying whether the model properly reproduced the biological motor control. Specifically, they claim that the model reproduced ataxic reaching which resemble to the cerebellum patients. However, it is not surprising that a poorly tuned feedback controller (i.e. PID controller) shows an oscillation. Therefore, it is necessary for the authors to show that their model significantly resembles the biological motor control by comparing to the model which shares the hierarchical structures but the error is corrected within the cortex and only motor commands descend to the spinal cord.

2) This model has attributed all of the learning effects to the synaptic connection between the cortex and spinal cord (descending and ascending connections). However, it has been repeatedly demonstrated that plastic changes within the cortex occur during motor learning [1 and others]. Therefore, the physiological relevance of attributing everything to descending and ascending pathways is questionable.

[1] Roth, R. H. et al. Cortical Synaptic AMPA Receptor Plasticity during Motor Learning. Neuron 105, 895-908.e5 (2020).

3) Relating to the previous comment, this model only changed the connection with the spinal cord, but the equivalent circuit can be achieved by changing the intracortical synaptic connection. It is necessary to justify why the connection with the spinal cord should be the sole target of learning.

4) The authors claim that previously known phenomena (convergent force field and rotational dynamics) "emerge" with this learning model. However, this is not supported by the results. Firstly, when they tested the convergent force field, the spinal cord is separated from the descending and ascending connections with cortex. This means that this observed property is independent of the learning of the cortico-spinal connections. In addition, rotational dynamics is almost exclusively observed in the one-to-one cortical coupling, which the authors call a "less-plausible model. Therefore, these phenomena are not emergent properties as a result of their learning model. It is necessary to specify what are the essential factors for each observed phenomenon.

5) Regarding the synergistic model, their model has spinal interneurons recruiting two motoneuron pools, which is very different from the muscle synergies observed in animals

[2]. It is necessary to justify why this type of synergies should be used here.

[2] d'Avella A, Portone A, Fernandez L, Lacquaniti F (2006) Control of fast-reaching movements by muscle synergy combinations. The Journal of neuroscience: the official journal of the Society for Neuroscience 26:7791-7810

[Editors’ note: further revisions were suggested prior to acceptance, as described below.]

Thank you for resubmitting your work entitled "Self-configuring feedback loops for sensorimotor control" for further consideration by *eLife*. Your revised article has been evaluated by Tamar Makin (Senior Editor) and a Reviewing Editor.

The manuscript has been improved but there are some remaining issues that need to be addressed, as outlined below:

Although the reviewers were satisfied by the new analyses, they were also unanimous in thinking that the manuscript is hard to follow and that this complexity could make it less broadly appealing. Our recommendation is thus for the authors to try to simplify the text, minimising the mental burden to the reader. The main concerns were:

1) The five different "configurations" are not well explained/motivated throughout the paper.

2) Figure captions are too short, which makes it hard to understand the results (e.g., what is Figure 8 about?).

3) There are too many abbreviations, which again does not help comprehend the text.

4) The Discussion is hard to follow, foremost because it requires that the readers remember all five configurations, which were presented in the much earlier -and very long- Results section.

5) The overall logic could be potentially streamlined so the study becomes easier to understand.

In addition, you may want to consider addressing the comment by Reviewer #4, although it is not mandatory.

*Reviewer #4 (Recommendations for the authors):*

I really appreciate the authors revising the manuscript. The argument is now much clearer. I found that the authors are very open about what neural implementation this motor learning is occurring in, and it seems too early to suggest the neural implementation since the same learning can occur either in the spinal cord or in the cerebral cortex, or perhaps in other structures such as the cerebellum which is omitted by this model.

As a neurophysiologist, I am curious about what insights this computational model will provide into our understanding of the neural mechanism of motor learning. I understand that the key assumptions of this model are that (1) errors vary monotonically with motor command and (2) differential Hebbian learning at the output connections reduces the errors. The second assumption has already been discussed for possible implementation in the spinal cord. However, the validity of the first assumption is not yet clear. I would like to suggest the authors elaborate on what neural mechanism could transform errors in the multiple sensory systems (e.g visual systems) into the monotonic error such as muscle length. It is also important to mention where the desired somatosensory activity (Sp) comes from.

---

## [Author Response]

Essential revisions:1. The paper focuses on an intriguing new hypothesis: that the spinal cord minimizes an error signal generated in motor cortex. Given the novelty of this hypothesis and its implications for the present manuscript, it is critical that the authors provide additional controls/analyses that specifically address it. These include:a) What happens if error minimisation happened in the motor cortex rather than in the spinal cord; would the model still produce reaching movements? Do the cortical neurons look like actual neurons (distribution of directional tuning, rotational dynamics)? Do "synergies" still emerge in the spinal cord?b) Does a model with the same hierarchical structure that doesn't perform error minimisation and is still capable of reaching exhibit the properties that the current model does (in terms of motor cortical activity, synergies, etc)?

a) We have explicitly modeled the scenario where the input-output structure of the controller is resolved in motor cortex. We now emphasize that the results of the paper do not depend on whether this happens in motor cortex or spinal cord, or even on the learning rule. The main ingredients are properly configured feedback control, and a minimally realistic model of the sensorimotor loop. To leave no ambiguity about this point, we include a version of our model with no plasticity (configuration 3).

b) All the model configurations that we present in the paper show all experimental phenomena. Hopefully this will clarify this point.

2. The authors posit that motor cortex reports and error that is being minimised during learning. Under this assumption, the motor cortical signals should be reduced to zero, which is what indeed seems to happen, e.g., in Figure 2E. Nevertheless, it is known that the activity of motor cortical neurons is not zero during the execution of a correct movement, even in highly trained individuals: e.g., trained monkeys have rotation dynamics when reaching (Churchland Cunningham et al. Nature 2012) or hand cycling (Russo et al. Neuron 2018). How can these observations be reconciled with the current model?

The error being minimized does not entail that motor cortex activity disappears. We expand on this point on our reply to reviewer 2.

3. The cerebellum is mentioned a couple of times but not included in the model. Based on the known fact that the cerebellum receives and integrates both massive corticopontine input and spinocerebellar proprioceptive input, its role is hence often assumed to be exactly what the motor cortex does in this model. The rationale behind not letting Cerebellum assume this role, but instead making the motor cortex assume this role, is not clear.

The role assigned to learning in this model, and the role typically ascribed to the cerebellum are not actually the same.

Please see our reply to reviewer 2.

4. The authors claim that previously known phenomena (convergent force fields and rotational dynamics) "emerge" with their learning model. However, this is not supported by the results. Firstly, when they tested the convergent force field, the spinal cord was separated from the descending and ascending connections with cortex. This means that this observed property is independent of having learned the cortico-spinal connections. In addition, rotational dynamics were almost exclusively observed in the one-to-one cortical coupling, which the authors call a "less-plausible model. Therefore, these phenomena are not an emergent property that results from their learning model. It is necessary to specify what are the essential factors for each observed phenomenon.

We did not intend to claim that the phenomena in the paper arise from learning, and we hope that the current manuscript is clear about this point. In the case of an isolated spinal cord the synapses that learn are not present, so the phenomena is clearly an effect of the musculoskeletal model, and of the structure in the spinal cord circuit. We also try to make clearer that the rotational dynamics emerge for all the included configurations of the model.

5. This model has attributed all of the learning effects to the synaptic connection between the cortex and spinal cord (descending and ascending connections). However, it has been repeatedly demonstrated that plastic changes within the cortex occur during motor learning (Roth et al. Neuron 2020). How is the physiological relevance of attributing all the learning to descending and ascending pathways justified?

As clarified in our responses to reviewers 2 and 3, we do not claim that all of the learning is being done in spinal cord. We expect many types of learning to happen along the sensorimotor loop; our model is only addressing how the structure of the feedback controller is obtained. Moreover, we remain open to this learning happening in the motor cortex.

6. In relation to the previous comment: an equivalent circuit could be achieved by changing the intracortical synaptic connections. Why is the corticospinal connection the sole target of learning, and how do the results change (or remain unchanged) if learning happens in other parts of the circuit?

We have addressed this by introducing the version of our model where learning happens in the connections from SPA to M (configuration 2).

7. No model can reproduce the whole CNS. However, the reviewers think that a few key features should be added to the model:a) The authors have not included any form of feedback in the spinal cord. Indeed, short-latency stretch reflexes are critical for movement: coupled with motoneurons, they are the basic units to generate movement (Xu, Tianqui et al. PNAS 2018); models of human walking show that gait can emerge only considering sensory inputs to the cord (Song and Geyer J Physiol 2015); and their absence prevents recovery from spinal cord injury (Takeoka, Aya et al. Cell 2014). Ia afferents have perhaps the strongest monosynaptic glutamatergic synaptic input to spinal motoneurons and constitute a primary driver of spinal motoneuron excitability which this model is currently missing. Does adding short-latency feedback change the conclusion of the paper?

We have now included feedback from the muscle afferents to the spinal cord.

b) As the authors discussed in the manuscript, the model omits many components that are known to be involved in learning and, as a result, the model reproduces the impaired motor control rather than the intact state. However, this prevents us from verifying whether the model properly reproduced the biological motor control. Specifically, they claim that the model reproduced ataxic reaching which resemble to the cerebellum patients. However, it is not surprising that a poorly tuned feedback controller (i.e. PID controller) shows an oscillation. Therefore, it is necessary for the authors to show that their model significantly resembles the biological motor control by comparing to the model which share the hierarchical structures but the error is corrected within the cortex and only motor commands descend to the spinal cord.

As commented to reviewer 3, we hopefully have gone beyond showing that a feedback controller shows an oscillation when explaining common points between the model and qualitative traits of ataxic movements.

c) 20 % of corticospinal projections to motoneurons innervating the arm and particularly the hand are thought to be monosynaptic. Does their addition change the present results?

We have included projections with plastic synapses from motor cortex to the α motoneurons to illustrate that this does not change the results.

d) The proportions and the connectivity of the network make the model seem less biologically plausible, which should at least be addressed in the text. First, the number of neurons in the populations at various levels seems rather small compared to the immense size of these structures. Why not have more neurons in the networks? This applies to all parts of the model, also to the spinal network. In many species, the ratio of motor neurons to interneurons is about 1:10, yet in this model, it is more like 1:2 (the trio). Also, the connectivity has a bias towards the feedforward network, yet local recurrent connectivity seems to be widely present in the nervous system. The proportions in the connectivity from M to C could also be problematic. All the spinal units (c = [c1, …. ,cN]) receive commands/error signal from M (e = [e1, …. ,eM]). Is M>N? Presuming M is > N this would represent a converging input from M to C, and that is problematic since the corticospinal projections are rather sparse. More likely there is diverging connectivity from M to C. It is unclear to me if this is an important problem, it may not be, but it would be appropriate to address such proportions in the text.

We address these limitations in the paper, and in our response to reviewer 2.

e) The model does not include any form of "pattern generator", a limitation the authors are aware of, and denote as the "supraspinal pattern formation" problem. They seem to export this to another area of the brain while the motor cortex-spinal cord loop takes care of an appropriate execution. This needs to be clarified and/or addressed with the model.

As clarified in the paper, generation of desired sensory perceptions is beyond the scope of this paper, but an approach is outlined in our previous work (reference 11).

f) It is well-known that the spinal cord autonomously can produce motor behaviors. It is unclear whether C in the model is capable of this.

We show that an isolated spinal cord has convergent force fields, which would produce a rudimentary type of reaching. Beyond this we don’t believe our spinal cord model can do much.

g) Regarding the synergistic model, their model has spinal interneurons recruiting two motoneuron pools, which is very different from the muscle synergies observed in animals (d'Avella et al., J Neurosci 2006). It is necessary to justify why this type of synergies should be used here.

As explained to reviewer 3, we used a natural proof of principle for the viability of Motor Synergy Encoders (reference 56). Whether the model can express other type of synergies is an interesting avenue for future research.

Reviewer #1 (Recommendations for the authors):The authors aimed to study corticospinal control of reach movements. More specifically they aim to understand how an unsupervised system could learn to reach and whether that system would then show properties observed in experiments. For this, the authors conceived a neural network model with defined hierarchy and architecture coupled to a simple biomechanical arm model in a closed-loop that is capable of learning a reach-out task. Interestingly, the model shows several properties found in experimental data such as directional tuning of motor cortical units as well as discrete force fields in the spinal cord.StrengthsThe paper has many strengths. It is rigorous and well written, and while not well articulated in the introduction I completely agree with the authors that computational models are necessary if we want to aim at any variation of the concept of "understanding" in relation to the central nervous system. Therefore their goal is absolutely significant and relevant.I also particularly liked the approach to start from specific hypotheses on the structure and function of the different modeled layers while leaving completely free the organization and strength of synapses in the system. This approach is elegant because starting from clear hypotheses it allows to observe what properties emerge "spontaneously" and compare these properties with experimental data.The results obtained are reasonable in terms of kinematics and it's interesting to observe the emergence of these properties from an error minimization rule imposed within the spinal cord.WeaknessesThe conclusions that the authors make in the paper are very strong. Given the emergence of certain interesting properties within the nervous system, it is implicit the implication that this model can have some sort of general meaning, and most importantly, that it validates from the theoretical point of view the homeostatic control idea. More specifically, the paper starts with a very strong hypothesis, which is that the spinal cord is minimizing an error signal generated in motor cortex. For an experimental neuroscientist (as myself) this is intriguing but also strikes as a very strong interpretation of the role of what we call motor cortex that for decades has been imagined as an actuator of movements. Because of the importance of the implication I don't think that the properties shown in the paper are sufficient to make such a claim, even if implicit. Therefore I think that some form of control is required. I understand completely that models are hard to build, but I can't escape wondering what would happen if the motor cortex would not be producing an error signal. What if that error minimization happened instead in the motor cortex and not in the spinal cord? Would your model still produce reach movements? Would those cortical neurons show directionality? Would "synergies" still emerge in the cord? I think this is an important point if we're claiming that those properties emerge as a consequence of the fact that the motor cortex is calculating an error and the spinal cord is minimizing that error. Therefore, to validate the main hypothesis of the paper, which is that the motor cortex is calculating an error and the spinal cord is minimizing that error, then I would like to see that a model that does not do that, but it's still capable of reaching while maintaining the same hierarchical architecture would not show the properties that the authors found in their model.

Thank you for the various observations up to this point. The question of motor cortex as a generator of errors or as an actuator is a subtle one. We want to raise the possibility that the spinal cord could be an actuator, but as we make now clear, this is not the central theme of the paper. We have restructured the paper to clarify this.

The paper makes two main points: (1) feedback control with a realistic enough musculoskeletal/neural model can explain many experimental phenomena, and (2) This feedback control can be adaptively configured through differential Hebbian learning rules. To leave no room for misunderstanding we added two new configurations. One of these has no learning at all, it is just a feedback controller (configuration 3). The other one (configuration 2) has the learning in motor cortex rather than the spinal cord. All the results hold for both of these configurations.

The second weakness is somehow related to the first. There are certain assumptions on the structure that are not true experimentally and it would be important to incorporate these because they may change the observed properties and could thus be used as further validation.The first missing property is direct cortico-spinal control of spinal motoneurons. The authors say in the introduction that the motor cortex does not activate spinal motoneurons directly and only connects to interneurons. Instead, the major difference between primates (including humans) and all the other animals is precisely the existence of direct monosynaptic connections between the motor cortex and muscles of the upper limb (particularly the hand). These monosynaptic components are thought to represent about 20% of the cortico-spinal tract. It would be important to see whether their presence changes the results that they obtained.

We did not mean to convey that direct motoneuron connections are entirely absent. The point that we want to convey is that the most dominant form of connectivity is through interneuron connections. As far as we know, direct connections are found in advanced primates, and mostly for distal joints, accounting more for manual dexterity than for reaching.

The good news is that our learning rule works well after introducing these direct connections, which are now part of the model.

The second more important functional inaccuracy concerns the spinal cord. The authors send their sensory signals to the "thalamus" which then relays them to the "sensory cortex" and "motor cortex". In the supplementary material they described the fact that they decided not to include what they call "short reflex" that would connect "the thalamus" to the spinal cord, or provide a feedback in some way. Therefore in their model, the spinal cord is not receiving a feedback at all. Unfortunately, this is in stark contrast with the actual structure of the spinal cord. Proprioceptive afferents are pivotal members of the spinal motor infrastructure. They are so important than in simple animals, coupled with motoneurons, they constitute the units that generate movement (Xu, Tianqi, et al. "Descending pathway facilitates undulatory wave propagation in *Caenorhabditis elegans* through gap junctions." Proceedings of the National Academy of Sciences 115.19 (2018): E4493-E4502.); models of human locomotion show that it is possible to build walking models without pattern generators that only consider sensory inputs int he spinal cord (Song, S., and Geyer, H. (2015). A neural circuitry that emphasizes spinal feedback generates diverse behaviours of human locomotion. J. Physiol. 593, 3493-3511.) and they seem to be critical to direct motor learning and recovery after spinal cord injury (Takeoka, Aya, et al. "Muscle spindle feedback directs locomotor recovery and circuit reorganization after spinal cord injury." Cell 159.7 (2014): 1626-1639.) Most importantly Ia afferents have perhaps the strongest monosynaptic glutamatergic synaptic input to the spinal motoneurons and constitute a primary driver of spinal motoneuron excitability which this model is currently missing. Additionally, some of the agonist-antagonist relationship that the authors find could be significantly changed by the presence of these afferents because Ia afferents connect not only to homologous motoneurons but up to 60% of motoneurons of synergistic muscles and they inhibit antagonists via Ia inhibitory interneurons (See Moraud E. (2016) Mechanisms Underlying the Neuromodulation of Spinal Circuits for Correcting Gait and Balance Deficits after Spinal Cord Injury, Neuron).In consequence, the presence of this "short-feedback" seems paramount to me to have an accurate representation of the spinal cord, and it's important to verify that this structure would not change the main conclusions of the paper.Indeed, as the authors have correctly implied, the main difference between the nervous system and an artificial neural network is the underlying defined architecture and hierarchy that must be represented at least for the most important elements. And spinal sensory feedback is a building block of movement and learning. Indeed, completely paralyzed rats are capable of learning different walking patterns when totally disconnected from the brain if excitation is provided by means of electrical stimulation and likely sensory afferents are the driver of this learning (Courtine, Grégoire, et al., "Transformation of nonfunctional spinal circuits into functional states after the loss of brain input." Nature neuroscience 12.10 (2009): 1333-1342.).

This is a very important observation. We agree that there is an important role to be played by the connections from muscle afferents to interneurons. To address this omission we have included these connections, under the hypothesis that they play a role similar to connections from afferents to motor cortex, namely, predictive error reduction.

Reviewer #2 (Recommendations for the authors):In this study, the authors investigate how the motor cortex and spinal cord interact in a long-loop reflex organization, from a computational modeling perspective. The motor cortex represents kinematic aspects, i.e. the actual movement of the hand and arm, as opposed to dynamics, which would involve the forces exerted by the muscles or higher-level parameters of the movement. Nevertheless, to achieve the end goal of a certain movement of the hand, a transformation from actuator (muscle) coordinates has to take place, and this is a challenging process. The authors propose that the motor cortex report an error between the desired and the actual movement, use computational modeling to explain how a flexible error reduction mechanism can be achieved and explained by computational modeling of the interplay between cortex and the spinal cord.Strengths: A theoretical analysis of the interplay between the spinal cord and motor cortex has rarely been studied previously especially using the long-loop reflex, dynamics, pattern formation, and plasticity. This theoretical contribution is important, especially in the light of the vast new experimental data that is being generated by e.g. calcium imaging and electrophysiology and the new observations of rotational cortical dynamics. It is also rare that the role of the spinal cord is being incorporated into the executing of movement in parallel with the motor cortex.Weaknesses: Some of the assumptions hinges on a flexible error reduction mechanism, which is described in a previous (unpublished) study, which seems somewhat fragile. The model does not really get to the root of the problem of how the nervous system generates the desired movement, which the authors are aware of and call the "supra spinal pattern formation" problem. They seem to export this problem to another area of the brain and describe how the motor cortex-spinal cord loop takes care of an appropriate execution.The study evaluates multiple aspects of the model in comparison with experimental observation such as direction tuning curve, rotational dynamics, and learning.Overall, this is an interesting and well-executed study.

Thank you for your kind comments. Regarding the weaknesses of the study, I would like to reiterate my initial reply to the reviewer 1: a central theme of this is study is the explanatory power arising from feedback control and a full model of the sensorimotor loop. This is independent of the learning mechanism. Also, our previous study is now published (reference 11 in the manuscript).

Specific comments:There are a couple of issues regarding the model that is not so clear (or perhaps well-hidden somewhere in the supplement). It is stated that the role of the motor cortex is to report the "error signal" of the movement, i.e. the signal coming from SPA, which is the difference between the desired movement (SP) and the actual movement (SA). First, if the motor cortex is reporting the error, and this error is being minimized while learning, then the signal in the motor cortex should ideally be reduced to zero, which is also what seems to happen in e.g. Figure 2E. Nevertheless, it is known that activity in the motor cortex is not zero during the execution of correct movement, especially in highly trained individuals, e.g. trained monkeys have rotational dynamics when hand-cycling (Churchland 2012). This also seems to be the case in the current study with directional tuning (Figure 5) and rotation, which appear contradictory to me. This is a bit unclear how this is possible. Second, the cerebellum is mentioned a couple of times but not included in the model. It is known that the cerebellum receives and integrates both massive corticopontine input and spinocerebellar proprioceptive input and the role of the cerebellum is hence often assumed to do exactly what the motor cortex does in this model. The rationale behind not letting Cerebellum assume this role, but instead making the motor cortex assume this role, is not clear to me.

We did not wish to convey the idea that the role of motor cortex is to convey an error, serving as some sort of rely station, and we apologize if that’s the case. Surely, there is more interesting learning being done in motor cortex, and we try to remain open to that possibility. This is one of the reasons for having a configuration of the model where motor cortex neurons are driven by more than one error signal. This can also observed in the fact that in our model motor cortex mixes error and afferent signals to produce a simple form of predictive control.

In the current manuscript we try to clarify that even if the error is reduced to zero the activity in motor cortex does not disappear. Activity in motor cortex can be very high, but if excitation and inhibition are balanced in the spinal cord movement will not be generated. Since there are “dual” subpopulations in M, high homogeneous activity may equally activate excitatory and inhibitory interneurons. A careful look at figures 2, and S1-S4, S8, S9 reveals that the M activity is far from ever being zero, and considering that the firing rate units represent averages over a population, some individual neurons could have very high activity in an equivalent spiking model. For all configurations the lowest stationary activity in M oscillates around 0.2 (figure S4).

The results with directional tuning and rotation come from the fact that large errors, such as those happening after the presentation of a new target (panel D, figures 2, S1-S4), cause a sudden increase in activity from certain M units. This activity quickly returns to baseline level, as the error is reduced.

We believe that the role being ascribed to learning in this system, and the role typically assumed from cerebellum have some subtle distinctions. In our model learning is meant to determine the input-output structure of the system: which muscle contracts in response to an error. For illustration purposes we may think of the classic vestibulo-ocular reflex. Rotation of the head in one direction causes rotation of the eyeball in the opposite direction to stabilize gaze. The inputoutput structure of this system is determined: the head and the eyeball rotate in opposite directions. The degree of this reflex rotation, however, is adaptive, and the cerebellum plays a role here. It is generally thought that the cerebellum creates an internal model of the system being controlled, permitting model-predictive control, and thus increasing performance of the controller. Most cerebellar models following this paradigm assume that a feedback control system with a good-enough configuration is already in place.

The current model does not explain what the authors call the "supra spinal pattern formation" problem, i.e. how the motor commands arise in the neuronal networks in the cortex and elsewhere. The problem of motor commands is conveniently exported to the external parameter "SP" module that is generating the intended movement. This is for a good reason since it is indeed a hard problem to explain how motor programs arise. I do not see a solution to this issue, only that it is a limitation to be aware of, which the authors seem to be. Nevertheless, the model proposed by the authors makes an effort to explain some of the dynamics of the different regions involved in properly learning and executing the motor commands.

This is indeed a difficult problem. We believe that “exporting” the problem to finding a desired set of perceptions “SP” offers an advantage. First, it partially solves the part of creating the spatiotemporal pattern of neuronal activity producing meaningful movement: this arises from the feedback control. Secondly, we believe that finding appropriate “SP” patterns is a task that could be solved with the right hierarchical architecture. More details are in our previous paper (reference 11).

I have some concerns regarding the network, the proportions, and the connectivity that make the model seem less biologically plausible, which should at least be addressed in the text. First, the number of neurons in the populations at various levels seems rather small compared to the immense size of these structures. Why not have more neurons in the networks? This applies to all parts of the model, also to the spinal network. In many species, the ratio of motor neurons to interneurons is about 1:10, yet in this model, it is more like 1:2 (the trio). Also, the connectivity has a bias towards the feedforward network, yet local recurrent connectivity seems to be widely present in the nervous system. The proportions in the connectivity from M to C could also be problematic. All the spinal units (c = [c1, …. ,cN]) receive commands/error signal from M (e = [e1, …. ,eM]). Is M>N? Presuming M is > N this would represent a converging input from M to C, and that is problematic since the corticospinal projections are rather sparse. More likely there is diverging connectivity from M to C. It is unclear to me if this is an important problem, it may not be, but it would be appropriate to address such proportions in the text.

The reviewer is correct that there is a vast gap between the numbers (and proportions) in the real brain, and those in our model. Despite being a “minimal” model, our codebase is rather large, and simulation times are borderline problematic. Simulations including time delays, multiple regions (in a closed loop with a physical plant), and multiple forms of plasticity tend to require care. We don’t include more neurons because we don’t want to increase complexity without a strong motivation. Given that our firing rate units represent the average activity of a population, for a large number of units the net effect of different population sizes can be reflected by the strength of the connectivity, unless there are significant dynamics beyond those provided by the E-I circuit. Still this is a limitation of the model, and we expose it in the paper (line 1634).

This observation also applies to our M to C connections. Interestingly, our simulations with plasticity in M to C begin with random full connectivity, but only have a few strong connections at the end. This result is not presented to avoid further bloating of the paper.

There is indeed a bias towards the feedforward network. We also include ascending connections from A to C and M (creating three separate loops), and lateral connections in M, SPA, and C. Our lateral connections in cortex have the role of supporting the emergence of “dual” representations. Surely there are more subtle roles for recurrent and lateral connections, but in this paper we remain agnostic about them. Once more, we don’t introduce elements unrelated to the goals of the model.

Reviewer #3 (Recommendations for the authors):This study proposes a new computational model for learning upper limb reaching control. In particular, this study develops a model that satisfies biological plausibility, which has not always been satisfied by previous models: (1) a network of the whole sensorimotor loop, (2) a model using neural elements, (3) learning using local information, and (4) continuous online learning. Specifically, their model showed that learning in the synaptic weights of descending and ascending projections between the cortex and spinal cord can reproduce ataxic reaching often observed in cerebellar patients. Furthermore, various phenomena that have been demonstrated in the motor control field emerge with this model. Based on these results, the authors argue that this is a biologically plausible learning model for animals' reaching control.Strength:1) This study examines a number of previous motor learning models and defines the constraints to develop a biologically plausible model. Given that there are too many parameters to be considered to build the motor control model, I truly appreciate the authors defining the constraints for a biological motor learning model and clarifying the research goal.Weakness:1) As the authors discussed in the manuscript, the model omits many components that are known to be involved in learning and, as a result, the model reproduces the impaired motor control rather than the intact state. However, this prevents us from verifying whether the model properly reproduced the biological motor control. Specifically, they claim that the model reproduced ataxic reaching which resemble to the cerebellum patients. However, it is not surprising that a poorly tuned feedback controller (i.e. PID controller) shows an oscillation. Therefore, it is necessary for the authors to show that their model significantly resembles the biological motor control by comparing to the model which shares the hierarchical structures but the error is corrected within the cortex and only motor commands descend to the spinal cord.

The qualitative characteristics of reaching in cerebellar patients have been studied for many years. It is indeed natural to think of tremors as faulty feedback control; Norbert Wiener (of Cybernetics fame) and many others were already thinking this way decades ago. We tried to go beyond this by: (1) the use of a closed-loop neuromusculoskeletal model, and (2) compiling and comparing the (somewhat sparse) ataxia literature relevant to our testing conditions.

A main point of the paper (emphasized in this new submission) is that feedback control is a powerful hypothesis, often forgotten by people explaining phenomena in motor control. We now include a model where the error is corrected within cortex, but the point still remains.

2) This model has attributed all of the learning effects to the synaptic connection between the cortex and spinal cord (descending and ascending connections). However, it has been repeatedly demonstrated that plastic changes within the cortex occur during motor learning [1 and others]. Therefore, the physiological relevance of attributing everything to descending and ascending pathways is questionable.[1] Roth, R. H. et al. Cortical Synaptic AMPA Receptor Plasticity during Motor Learning. Neuron 105, 895-908.e5 (2020).

We did not intent to attribute everything to ascending and descending pathways. As mentioned to reviewer 2, we remain open to other type of learning in motor cortex, and as should be evident in this new version of the paper, we also remain open to learning configuration of the feedback control in motor cortex.

3) Relating to the previous comment, this model only changed the connection with the spinal cord, but the equivalent circuit can be achieved by changing the intracortical synaptic connection. It is necessary to justify why the connection with the spinal cord should be the sole target of learning.

To address this issue we have introduced the configuration 2 of the model, which does precisely this.

4) The authors claim that previously known phenomena (convergent force field and rotational dynamics) "emerge" with this learning model. However, this is not supported by the results. Firstly, when they tested the convergent force field, the spinal cord is separated from the descending and ascending connections with cortex. This means that this observed property is independent of the learning of the cortico-spinal connections. In addition, rotational dynamics is almost exclusively observed in the one-to-one cortical coupling, which the authors call a "less-plausible model. Therefore, these phenomena are not emergent properties as a result of their learning model. It is necessary to specify what are the essential factors for each observed phenomenon.

We did not mean to attribute all results to learning in the cortico-spinal connections, and certainly not the result of convergent force fields, where those connections are not even present. We tested the convergent force fields with an isolated spinal cord because that is the experimental paradigm, and attributed the result to viscoelastic properties of the arm, forces/torques adding linearly, and the balance between agonists and antagonists.

Giving the impression that we attribute every result in the paper to synaptic learning is a fault of our manuscript. We have tried to be very clear about this with the inclusion of configuration 3, where no plasticity is present, but the results appear nonetheless.

The appearance of the plots showing rotational dynamics can change a lot depending on the hyperparamters of the model, and on the detail of the algorithm, such as the sampling window. These plots are different from those in Churchland et al. 2012: we have few conditions, few neurons, no overtrained model, and a different way to produce the plots. To assess the presence of quasi-oscillations I would advice to also pay attention to the numerical measure (percentage of variance in the first jPCA plane). Notice that we have twice as many jPCA planes as the original study, but close to the same variances.

5) Regarding the synergistic model, their model has spinal interneurons recruiting two motoneuron pools, which is very different from the muscle synergies observed in animals[2]. It is necessary to justify why this type of synergies should be used here.[2] d'Avella A, Portone A, Fernandez L, Lacquaniti F (2006) Control of fast-reaching movements by muscle synergy combinations. The Journal of neuroscience: the official journal of the Society for Neuroscience 26:7791-7810

The synergistic model is a proof of principle, stating that it is not incompatible with Motor Synergy Encoders (reference 56), and this could be an interesting avenue of future research.

The study in d’Avella et al., 2006 takes the electromyograph waveforms and applies an optimization algorithm to find about five time-dependent patterns (the “synergies”) that explain most of the variance (which turned out to be around 80%). This is indeed a different interpretation of synergies from the one we use. In the introduction we clarified that “synergy” has multiple meanings, but we focus on Motor Synergy Encoders. This is because they fit naturally with the model.

This, however, got me thinking (SVF). Who is to say that applying the optimization procedure in d’Avella et al. 2006 would not lead to explain most variance through a reduced number of waveforms? I think this would be a very interesting future experiment. Thank you for providing the reference, and leading me in this path of thought.

[Editors’ note: further revisions were suggested prior to acceptance, as described below.]

The manuscript has been improved but there are some remaining issues that need to be addressed, as outlined below:Although the reviewers were satisfied by the new analyses, they were also unanimous in thinking that the manuscript is hard to follow and that this complexity could make it less broadly appealing. Our recommendation is thus for the authors to try to simplify the text, minimising the mental burden to the reader. The main concerns were:1) The five different "configurations" are not well explained/motivated throughout the paper.

To reduce the confusion caused by the multiple configurations we have taken 3 steps. First, we have found that two of the configurations are not necessary to the main argument in the paper, so they have been relegated to the Appendix. Second, rather than using numbers, each configuration now has a descriptive name. Third, we have further clarified the rationale for each configuration.

2) Figure captions are too short, which makes it hard to understand the results (e.g., what is Figure 8 about?).

We wrote more descriptive figure captions.

3) There are too many abbreviations, which again does not help comprehend the text.

We removed some abbreviations, and sometimes repeated their definitions.

4) The Discussion is hard to follow, foremost because it requires that the readers remember all five configurations, which were presented in the much earlier -and very long- Results section.

The Discussion has been redacted, once more stating the idea behind each configuration.

5) The overall logic could be potentially streamlined so the study becomes easier to understand.

We have tried to improve the most tedious paragraphs.

In addition, you may want to consider addressing the comment by Reviewer #4, although it is not mandatory.Reviewer #4 (Recommendations for the authors):I really appreciate the authors revising the manuscript. The argument is now much clearer. I found that the authors are very open about what neural implementation this motor learning is occurring in, and it seems too early to suggest the neural implementation since the same learning can occur either in the spinal cord or in the cerebral cortex, or perhaps in other structures such as the cerebellum which is omitted by this model.As a neurophysiologist, I am curious about what insights this computational model will provide into our understanding of the neural mechanism of motor learning. I understand that the key assumptions of this model are that (1) errors vary monotonically with motor command and (2) differential Hebbian learning at the output connections reduces the errors. The second assumption has already been discussed for possible implementation in the spinal cord. However, the validity of the first assumption is not yet clear. I would like to suggest the authors elaborate on what neural mechanism could transform errors in the multiple sensory systems (e.g visual systems) into the monotonic error such as muscle length. It is also important to mention where the desired somatosensory activity (Sp) comes from.

We found the comment by Reviewer #4 to be very insightful. Dealing with monotonicity of the errors and with the origin of the S_P_ patterns is crucial to eventually extend this work into a more general system (for example, to include vision). Considering the current complexity of the manuscript, we believe it is best to refer the interested readers to our previous work (enclosed with the submission). In particular, our previous paper (first referenced in line 42) develops a proof-of concept for how an actor-critic system can be used to deal with non-monotonicity. We have also suggested (line 264) to consult that paper when considering the origin of the S_P_ patterns.